



# Fate of the nitrate radical at the summit of a semi-rural mountain site in Germany assessed with direct reactivity measurements

Patrick Dewald, Clara M. Nussbaumer, Jan Schuladen, Akima Ringsdorf, Achim Edtbauer, Horst Fischer, Jonathan Williams, Jos Lelieveld and John N. Crowley

Atmospheric Chemistry Department, Max Planck Institut für Chemie, 55128 Mainz, Germany

*Correspondence to*: John N. Crowley (john.crowley@mpic.de)

**Abstract.** The reactivity of $NO_3$ plays an important role in modifying the fate of reactive nitrogen species at nighttime. High reactivity (e.g. towards unsaturated VOCs) can lead to formation of organic nitrates and secondary organic aerosol, whereas

low reactivity opens the possibility of heterogeneous $NO_X$ losses via formation and uptake of $N_2O_5$ to particles.

We present direct $NO_3$ reactivity measurements ($k^{NO_3}$) that quantify the VOC-induced losses of $NO_3$ during the TO2021 campaign at the summit of the Kleiner Feldberg mountain (825 m, Germany) in July 2021. $k^{NO_3}$ was on average ~ 0.035 s$^{-1}$ during the daytime, ~ 0.015 s$^{-1}$ for almost half of the nights and below the detection limit of 0.006 s$^{-1}$ for the other half, which may be linked to sampling from above the nocturnal surface layer. $NO_3$ reactivities derived from VOC measurements and the

corresponding rate coefficient were in good agreement with $k^{NO_3}$, with monoterpenes representing 84 % of the total reactivity. The fractional contribution $F$ of $k^{NO_3}$ to the overall $NO_3$ loss rate (which includes additional reaction of $NO_3$ with NO and photolysis) were on average ~16 % during the daytime and ~50-60 % during the nighttime. The relatively low nighttime value of $F$ is related to the presence of several tens of pptv of NO on several nights. $NO_3$ mixing ratios were not measured but steady-state calculations resulted in nighttime values between < 1 pptv and 12 pptv. A comparison of results from TO2021 with direct

measurements of $NO_3$ during previous campaigns between 2008 and 2015 at this site revealed that $NO_3$ loss rates were remarkably high during TO2021, while $NO_3$ production rates were low.

We observed NO mixing ratios of up to 80 pptv at night which has implications for the cycling of reactive nitrogen at this site. With $O_3$ present at levels of mostly 25 to 60 ppbv, NO is oxidised to $NO_2$ on a time-scale of a few minutes. We find that to maintain NO mixing ratios of e.g. 40 pptv requires a ground-level NO emission rate of 0.33 pptv s$^{-1}$ (into a shallow surface

layer of 10 m depth). This in turn requires rapid deposition of $NO_2$ to the surface ($vd_{NO2}$ ~ 0.15 cm s$^{-1}$) to reduce nocturnal $NO_2$ levels to match the observations.

## 1 Introduction

Nitric oxide (NO) and nitrogen dioxide ($NO_2$) are atmospheric pollutants, which exert a great impact on climate and air quality (Pozzer et al., 2012; Lelieveld et al., 2020). As $NO_2$ is the source of boundary layer ozone ($O_3$, which is phytotoxic and a cause





of respiratory illness), understanding the processes that remove $NO_x$ (= NO + $NO_2$) are of great importance (Crutzen and Lelieveld, 2001; Lelieveld et al., 2016; Edwards et al., 2017). The formation of long-lived or soluble organic nitrates during the oxidation of volatile organic compounds (VOCs) provides a mechanism to convert $NO_X$ to $NO_Z$ (where $NO_z$ includes both organic and inorganic nitrates in the gas- and particle-phase), which may be transported away from the source region or removed via dry- or wet-deposition, respectively (Rollins et al., 2012; Present et al., 2020).

The major initiators of VOC oxidation are hydroxyl radicals (OH), ozone ($O_3$) and the nitrate radical ($NO_3$) (Ng et al., 2017; Wennberg et al., 2018) with OH reactions most important during the daytime (Lelieveld et al., 2008). The $NO_3$ radical is generally considered to be important only at nighttime (Brown and Stutz, 2012) although in some environments, it can also contribute substantially to the oxidation of unsaturated VOC during the day (Liebmann et al., 2018a; Liebmann et al., 2018b). $NO_3$ is formed almost exclusively in the sequential oxidation of NO by $O_3$ (R1 and R2). During daytime, $NO_3$ is lost via rapid

photolysis (R5 and R6, with a lifetime of seconds) and an efficient reaction with NO ($k_7$ = 2.6 x $10^{-11}$ $cm^3$ $molecule^{-1}$ $s^{-1}$ at 298 K) (IUPAC, 2022), which result in low mixing ratios (Wayne et al., 1991). $NO_3$ also reacts with $NO_2$ to form dinitrogen pentoxide ($N_2O_5$), which is in thermal equilibrium with $NO_3$ and $NO_2$ (R3, R4).

$$NO + O_3 \rightarrow NO_2 + O_2 \tag{R1}$$
$$NO_2 + O_3 \rightarrow NO_3 + O_2 \tag{R2}$$
$$NO_3 + NO_2 + M \rightarrow N_2O_5 + M \tag{R3}$$
$$N_2O_5 + M \rightarrow NO_3 + NO_2 + M \tag{R4}$$
$$NO_3 + hv \rightarrow NO + O_2 \tag{R5}$$
$$NO_3 + hv \rightarrow NO_2 + O \tag{R6}$$
$$NO_3 + NO \rightarrow 2NO_2 \tag{R7}$$

Reactions R1 - R4 can result in permanent loss of $NO_x$ from the gas phase through deposition or uptake to particles of e.g. $NO_3$ or $N_2O_5$ (R8, R9) (Crowley et al., 2011; Phillips et al., 2016).

$$N_2O_5 \rightarrow \text{particle (or deposition)} \tag{R8}$$
$$NO_3 \rightarrow \text{particle (or deposition)} \tag{R9}$$

In forested regions during the night, $NO_3$ reacts predominantly with unsaturated volatile organic compounds (VOC) often of

biogenic origin such as isoprene or monoterpenes, which results in the formation of alkyl nitrates ($RONO_2$) (R10) (Hallquist et al., 1999; Fry et al., 2014; Wu et al., 2021). Depending on the biogenic VOC involved, the $RONO_2$ formed may have low volatility and may deposit to surfaces or transfer to the particle phase to form secondary organic aerosols (SOA) (R11) (Place et al., 2022). The reaction between $NO_3$ and BVOC consequently represents a loss of $NO_X$ from the gas-phase and thus has an impact on air quality via suppression of ozone formation and increases in SOA levels (Fry et al., 2011; Romer Present et

al., 2020).

$$NO_3 + VOC (+ O_2) \rightarrow \rightarrow RONO2 \tag{R10}$$
$$RONO_2 \rightarrow \text{deposition/SOA} \tag{R11}$$



The nocturnal $NO_3$ lifetime close to the surface is generally short (typically in the range of minutes) owing to the build-up in concentration of reactive gases emitted from the biosphere into a shallow nocturnal boundary layer (Liebmann et al., 2018a;
Liebmann et al., 2018b). Longer $NO_3$ lifetimes (sometimes exceeding 1 hour) have been derived from $NO_3$ measurements in very clean regions (Allan et al., 2000; Martinez et al., 2000), from measurements in the overlying residual layer using towers and aircraft platforms (Stutz et al, 2004; Brown et al., 2007a; Brown 2007b) and at mountain sites where the meteorological situation results in the measurement location being above the nocturnal surface layer (Carslaw et al., 1997; Brown et al., 2016; Sobanski et al., 2016).

The lifetime of $NO_3$ has often been derived using a stationary-state approximation, which relies on direct measurements of $NO_3$, $NO_2$ and $O_3$ (Heintz et al., 1996; Allan et al., 1999; Geyer et al., 2001; Brown et al., 2004; Brown et al., 2009; Stutz et al., 2010; Sobanski et al., 2016). This method is limited to periods when $NO_3$ mixing ratios are above the instrumental detection limit, which (depending on instrument performance) may restrict the method to periods when $NO_3$ production rates are high and $NO_3$ reactivities (i.e. the inverse of $NO_3$ lifetimes) are low. This is usually not the case during the daytime or even during
the nighttime in areas with high BVOC emissions (Liebmann et al., 2018a). Direct $NO_3$ reactivity measurements not only extend the accessibility to daytime reactivities but also, together with measurements of NO, $NO_3$, photolysis rates ($J_{NO_3}$) and VOCs, enables the determination of the fate of the $NO_3$ radical throughout the diel cycle. Recent direct $NO_3$ reactivity measurements and model calculations (Liebmann et al., 2019; Foulds et al., 2021) suggest that $NO_3$ also contributes to daytime alkyl nitrate formation, which typically occurs through the OH-initiated oxidation of BVOC in the presence of NO (Wennberg
et al., 2018). Quantifying the contribution of $NO_3$ + VOC reactions to the $NO_3$ reactivity is thus central in understanding the role of $NO_3$, in e.g. SOA formation and $NO_X$ lifetimes.

In this study, the fate of the $NO_3$ radical on the semi-rural Kleiner Feldberg mountain (in the south-west of Germany) in July and August 2021 (TO2021 campaign) during both day- and nighttime is analysed by direct measurements of NO, photolysis rates ($J_{NO_3}$) and the first-order $NO_3$ loss-constant resulting from reaction with VOCs ($k^{NO_3}$). Measurements of VOCs that are
reactive towards $NO_3$ enable us to calculate their fractional contribution to $k^{NO_3}$. With the help of $NO_3$, $NO_2$, NO and $O_3$ measurements we derive $NO_3$ loss-terms via the steady-state assumption ($L_{NO_3}$) for previous campaigns at this site to assess the impact of differing meteorological and chemical conditions.

## 2 The TO2021 campaign

The TO2021 campaign took place in July and August 2021 at the Taunus Observatory (TO) at the summit of the Kleiner
Feldberg mountain (825 m above sea level). A detailed description of the location has been given elsewhere (Crowley et al., 2010) and only a brief summary is given here. The Kleiner Feldberg is mostly surrounded by coniferous forest, but an area at the summit (~ 100 m$^2$) is cleared of trees and hosts the meteorological measurements of the German Meteorological Service (Deutscher Wetterdienst, DWD) and permanent measurement containers of the University of Frankfurt and the Hessian Agency for Nature Conservation, Environment and Geology (Hessisches Landesamt für Naturschutz, Umwelt und Geologie,





HLNUG). The summit itself is covered with bushes and, especially to the north, with blueberry shrubs. The mountain tops of
      Altkönig (798 m a.s.l.) and Großer Feldberg (878 m a.s.l.) are in the direct vicinity (< 3 km). Air arriving from the south-west
      and south-east is impacted by anthropogenic emissions from the densely populated cities of Frankfurt, Wiesbaden and Mainz
      (20-30 km), whereas air from the north-west, north and north-east is cleaner, with no major cities for 50-70 km.

**2.1 Instrumentation**

For the duration of the TO2021 campaign, two (stacked) containers including the instruments operated by the Max-Planck-
      Institute for Chemistry (MPIC) were set up on the site. If not stated otherwise, the instruments sampled from a high-volume-
      flow stainless-steel tube ($10 \text{ m}^3 \text{ min}^{-1}$, 0.2 s residence time) sucking air from ca. 10 m above the ground. Each instrument with
      measurements used in the analysis is described below.

**2.1.1 NO₃ reactivity**

The Flow-Tube Cavity Ring Down Spectrometer (FT-CRDS) setup used to quantify VOC-induced $NO_3$ reactivity (Liebmann
      et al., 2017)  consists of a Teflon coated (FEPD 121, Chemours) glass flow-tube reactor, in which a flow of ambient air is
      mixed with 30-60 parts per trillion per volume (pptv) of synthetically generated $NO_3$.
      $NO_3$ is generated by the sequential oxidation of NO and $NO_2$ (3.5 sccm of 1 parts per million per volume (ppmv) in $N_2$, Air
      Liquide) by $O_3$ (generated by passing synthetic air over a Hg lamp) in an upstream Teflon-coated glass reactor (thermostated
to 30°C at a pressure of 1.3 bar) in 400 standard (STP) cubic centimetre per minute (sccm) synthetic air. The flow exiting the
      $NO_3$ source is passed through ca. 15 cm ¼ inch (in.) outer diameter (OD) PFA tubing that is heated to 140°C so that $N_2O_5$ is
      quantitatively decomposed to $NO_3$ and $NO_2$ (R4). The flow from the $NO_3$ source is then mixed with either 2800 sccm synthetic
      or ambient air and passed through the flow-tube reactor where it resides for time *t*. The synthetic air used to measure zero
      reactivities was provided by a commercial zero-air generator (CAP 120, Fuhr GmbH) and humidified to ambient level with a
permeation tube (PermaPure, MH-070-24F-4) immersed in deionized water. The ambient air was sampled from the high-flow
      inlet through ¼ in. (OD) PFA tubing equipped with a Teflon membrane filter (2 µm pore, 47 mm diameter, Pall Corp.).
      $NO_3$ surviving the flow-tube was detected by CRDS at 662 nm. The ring-down time in the absence of $NO_3$ was determined
      every ca. 5 min by adding an excess of NO (3 sccm of 100 ppmv in $N_2$). $NO_3$ reactivities are deduced from the relative change
      in $NO_3$ mixing ratio in ambient air compared to synthetic air. Dynamic dilution of the ambient air with synthetic air was used
to keep the $NO_3$ reactivity in a measureable range when sampling highly reactive air masses.
      Since the $NO_3$ mixing ratio is affected by reactions R1-R4, R7 and R9 in addition to the reaction of interest (R10), a numerical
      simulation procedure that corrects for the impact of NO and $NO_2$ is necessary to extract the $NO_3$ reactivity towards VOCs
      ($k^{NO_3}$). The validity of this correction procedure was checked by adding a known amount of NO (1-6 sccm of 245 parts per
      billion per volume (ppbv) NO in $N_2$, Air Liquide) every two hours during the zeroing periods throughout the campaign. As
shown in Fig. S1a of the Supplement, the model was able to reproduce the observed $NO_3$ mixing ratios reliably. A further
      calibration sequence during the campaign, in which five different amounts of NO were added, is displayed in Fig. S1b. The





flow-tube predominantly used during TO2021 features a residence time of $t = 9.5$ s and an $NO_3$ wall loss rate of 0.001 s⁻¹. The limit of detection (LOD) is mainly defined by the stability of the $NO_3$ source and baseline, which were improved by thermostating both the $NO_3$ source and the flowtube and insulating the cavity from thermal gradients in the container so that a
signal-stabilty related uncertainty of 16% was achieved. For the numerical simulation procedure, ambient $O_3$, NO and $NO_2$ mixing ratios and rate coefficients for (R1-R4, R7) were deployed. Liebmann et al. (2017) showed with the help of Monte Carlo simulations that the uncertainty associated with this simulation is dependent on the ratio between ambient $NO_2$ and $k^{NO_3}$. Assuming a typical daytime situation for TO2021 ($k^{NO_3} \sim 0.04$ s⁻¹, $[NO_2] = 2$ ppbv $NO_2$) the numerical simulation introduces an uncertainty of 15 %, resulting in an overall uncertainty of 22 %. However, if for example $k^{NO_3}$ is 0.006 s⁻¹ in the presence
of 1 ppbv $NO_2$ (as occasionally detected during the nighttime), the uncertainty caused by the simulation increases to ca. 50 %. During TO2021, the instrument's LOD was 0.006 s⁻¹ for this flowtube.

Between the 23ʳᵈ and 25ᵗʰ July, a larger flow-tube was tested with the intention of extending the LOD to lower reactivities. The residence time (20 s during the day or 32 s during the night according to position of a moveable injector) and wall loss rates were characterised during the campaign as detailed by Liebmann et al. (2017). The factor ~3 longer residence time at night
compared to the smaller flow-tube should have extended the LOD to 0.003 s⁻¹. However, the larger flow-tube suffered from a larger $NO_3$ wall loss rate (> 0.04 s⁻¹), which effectively worsened the LOD. For this reason, the deployment of this flowtube was stopped after two days.

During the nighttime, before being mixed with synthetic $NO_3$, the air was sampled through a 2 $L$ uncoated glass flask (40 s residence time) that was heated to 35°C. This ensures that ambient $NO_3$ and $N_2O_5$ does not reach the flow tube to bias the
measurement. The NO mixing ratios that were used in the numerical simulations were corrected (typically by a factor of 0.6) for the reaction with ambient $O_3$ during residence in the flask.

### 2.1.2 NO₂ , NO, O₃ and actinic flux

Owing to the importance of co-located $NO_2$ measurements for interpretation of the $k^{NO_3}$ data, the FT-CRDS set up has a second inlet and cavity to measure $NO_2$ (Liebmann et al., 2018b) with a total measurement uncertainty (defined by noise and
baseline stability) of 8 % and a LOD of 168 pptv (4 s). A further CRDS-based measurement of $NO_2$ was made using a thermal-dissociation cavity ring-down spectrometer (TD-CRDS) (Friedrich et al., 2020) for measurement of $NO_X$ and $NO_Y$. At nighttime, when NO was generally < 80 pptv, the $NO_X$ channel of this instrument essentially measures $NO_2$. The inlet of the $NO_X$ / $NO_Y$ instrument was located on the container roof, ~ 2 m to the north and 2 m lower than the top of the high-flow inlet. In addition, $NO_2$ was measured with a chemiluminescence (CLD) setup (ECO Physics, CLD 790 SR) equipped with a
photolytic converter to convert $NO_2$ to NO (Tadic et al., 2020; Nussbaumer et al., 2021). This instrument also provided the campaign NO data-set. Calibration (using a dynamically diluted, secondary 5 ppm NO standard) was carried out every 2 hours. The LODs for NO and $NO_2$ were 7 and 10 pptv, respectively, the total measurement uncertainties were 9 and 19 % for NO and $NO_2$.





The three sets of NO₂ measurements are compared in the Supplement (Fig. S2). A bivariate linear regression (York, 1966) of

the data sets yields offsets below the LOD of the FT-CRDS NO₂ cavity in both cases. An excellent agreement with the TD-CRDS measurement is observed (slope of 0.99), while a fair agreement (slope of 1.09) within associated uncertainties is achieved for the intercomparison with the CLD measurement. O₃ was measured via UV absorption with two identical, commercial ozone monitors (2B technologies, model 205) that were cross-calibrated after the campaign. The instrument background was estimated ca. every two days with synthetic air from the zero-air generator. The uncertainty associated with

this measurement is 5 % and the LOD is 2 ppbv.

Actinic flux measurements were made by a spectral radiometer (Metcon GmbH) installed on top of the upper container and converted to photolysis frequencies for NO₃ ($J^{NO_3}$) using evaluated absorption cross sections and quantum yields (Burkholder et al., 2016) with an overall uncertainty of ca. 15 % (Friedrich et al., 2021).

### 2.1.3 VOC measurements

VOCs were measured from the 15$^{th}$ to 31$^{st}$ July with a proton-transfer-reaction time-of-flight mass-spectrometer (PTR8000, IONICON Analytik GmbH) (Jordan et al., 2009; Bekö et al., 2020) with a time resolution of 20 s, operated with hydronium ions (H₃O⁺) at a pressure of 2.2 mbar and an E/N of 137 Td. Mixing ratios of isoprene, monoterpenes and sesquiterpenes are derived from calibrating to a gas standard containing isoprene, α-pinene and β-caryophyllene (Apel-Riemer Environmental Inc., Colorado, USA). The limit of detection lies in the range of tens of ppt and the uncertainty is defined to be below 20 %.

A second PTR-ToF-MS (VOCUS, Tofwerk AG) provided uncalibrated VOC data for the period between 20$^{th}$ July and 6$^{th}$ August (Krechmer et al., 2018). In order to extend data availability, the VOCUS data for isoprene, monoterpenes and sesquiterpenes were scaled to that of the PTR8000 data set during the common time period.

Both PTR-ToF-MS were located in a permanent container of the TO, ca. 8 m distant from the MPIC container. Air was sampled from the roof of the container (ca. 8 m) through a heated inlet line equipped with a polytetrafluoroethylene (PTFE) filter.

### 180 2.1.4 Temperature and relative humidity profiles

Deployment of a drone (EVO-X12, multikopter.de) equipped with a commercial gas sensor (BME680, Bosch Sensortech GmbH) enabled the measurement of vertical profiles of pressure, temperature and relative humidity (time resolution of 1 s) to a height of 100 m AGL.

### 3 Results and Discussion

An overview of the key meteorological and trace-gas measurements used in the analysis for the TO2021 campaign period from July to August 2021 is given in Fig. 1. Grey shaded areas mark the nighttime periods; sunrise during the measurement period was at ~ 03:30 and sunset at ~ 19:30 UTC. $k^{NO_3}$ shows a distinct daytime to nighttime variability and generally follows the





summed mixing ratio of monoterpenes (ΣMTs) which were present at maximum mixing ratios (during the day) of typically between 150 and 400 pptv.

Wind speeds were predominantly between 2 and 4 m s⁻¹ with most wind-sectors represented, although wind from the east and south-east originating from the Frankfurt area (SE) were rarely encountered. The local wind-directions and speeds during TO2021 are displayed as a wind rose in Fig. S3a in the Supplement.

There were several periods of rain and fog during TO2021, which is reflected by high relative humidities (RH) mostly between 75 and 100 % at moderate temperatures between 12 and 20°C. Ozone mixing ratios varied between 20 and 60 ppbv. The CLD

setup observed NO mixing ratios close to (10 to 20 pptv) or below the LOD of 7 pptv on ca. half of the nights, but also returned values of between 20 to 80 pptv for prolonged periods on some nights. Daytime NO mixing ratios were between 0.5 ppbv and 2 ppbv, with maximum values around midday. Spikes in NO mixing ratios caused by vehicles at the site were removed from the dataset. $NO_2$ mixing ratios (as measured with the FT-CRDS setup) were generally between 1 and 2 ppbv, with occasional values of up to 6 ppbv. Photolysis rates of $NO_3$ ($J_{NO_3}$) of ca. 0.15 s⁻¹ were detected at noon. The data-gap between the 3rd and

5th July was caused by a power-failure.

## 3.1 NO₃ reactivity

As is evident from Fig. 1, $k^{NO_3}$ followed the trend in monoterpene mixing ratios and was generally higher during the daytime compared to the night. As illustrated in a wind rose in the Supplement (Fig. S3b), $k^{NO_3}$ displayed no clear dependence on wind directions. A closer examination of the data reveals that the nights can be roughly divided into two types: On 15 of the 34

nights, NO₃ reactivities remained well above the instrument's LOD of 0.006 s⁻¹ (from now on defined as "Type-1" nights), whereas during 14 nights $k^{NO_3}$ was predominantly lower than 0.006 s⁻¹ ("Type-2"). The other 5 nights showed a transitional behaviour between those two types.

An example of a Type-1 night is shown in Fig. 2a. Following a late evening value of $k^{NO_3}$ ~ 0.1 s⁻¹ the NO₃ reactivity decreased during darkness from 0.08 s⁻¹ at 20:00 UTC to 0.02 s⁻¹ at 03:00 UTC. During this period, northerly winds with speeds around

4 m s⁻¹ prevailed and the decrease in reactivity cannot be related to a change in air-mass origin. At the same time, we observed a decrease in temperature (~17 to ~13 °C) that was accompanied by an increase in the relative humidity (78 to 98%) and a quasi-continuous reduction in O₃ mixing ratios from ~35 to ~25 ppbv. Note that ca. 20-30 pptv of NO were detected during this night, implying that reaction R7 would represent a significant loss process for NO₃. A detailed discussion of this aspect follows in section 3.4.

Figure 2b shows an example of a Type-2 night with a sharp decrease of $k^{NO_3}$ from ~ 0.02 s⁻¹ just before sunset to below the LOD (0.006 s⁻¹) within the first hour after sunset. As for Type-1, there is no significant change in the wind direction. However, in contrast to the Type-1 example, after a slight increase just after sunset, O₃ was roughly constant and significantly higher throughout the night with NO below the detection limit during the entire night. In addition the temperature (14 ± 1 °C) and relative humidity (70 ± 5 %) were roughly constant, the latter significantly lower than for the Type-1 example.





Low NO$_3$ reactivities at nighttime (i.e. Type-2 nights) can result from low rate of emission of biogenic VOCs (e.g. owing to low temperatures) but can also be associated with strong vertical gradients, which effectively decouple ground level emissions from the air above. For the latter case, we are dealing with a shallow surface layer with its top below the inlet, so that air is sampled from the nocturnal boundary or residual layer (Brown and Stutz, 2012) in which the NO$_3$ lifetimes can be very long. This phenomenon has been reported for this and other mountain sites (Carslaw et al., 1997; Brown et al., 2016; Sobanski et

al., 2016; Liebmann et al., 2017; Liebmann et al., 2018b). Slow exchange between the surface layer and the residual layer can result in strong gradients in trace gases such as O$_3$, which undergoes dry-deposition in the surface layer but is long-lived in e.g. the residual layer. The situation for NO$_2$ is more complex as it may be formed from the O$_3$-induced oxidation of near-surface emissions of NO and also lost via (slow) reaction with O$_3$ and dry-deposition (Brown et al., 2003b; Stutz et al., 2004; Brown et al., 2007a).

Figure 3 displays the campaign-averaged diel cycles of $k^{NO_3}$ (along with O$_3$, RH, T, NO and MTs) classified according to Type-1 or Type-2 nights. $k^{NO_3}$ was on average around 0.015 s$^{-1}$, during Type-1 nights, with a daytime reactivity of 0.04 s$^{-1}$ (Fig. 3a). The observed orders of magnitude for $k^{NO_3}$ are consistent with the directly measured nighttime NO$_3$ reactivities ranging between < 0.005 and up to 0.06 s$^{-1}$ during three nights in July 2015 (NOTOMO campaign) with the same instrument (Liebmann et al., 2017).

By definition, the median nighttime reactivity for Type-2 nights is at the instrument's LOD, while the median daytime reactivities prior to Type-2 nights are very similar to those observed prior to Type-1 nights. The median diel cycles for O$_3$ (Fig. 3b) differ significantly for the two types: during Type-1 nights O$_3$ decreases continuously (consistent with previous observations on this site (Handisides, 2001)), while during Type-2 nights, O$_3$ mixing ratios remain fairly constant and higher. There are also significant differences in the median NO mixing ratio, with nightime values (Fig. 3f) mostly below or close (10-

12 pptv) to the LOD during Type-2 nights and values of 30-40 pptv during Type-1 nights.

The lower nighttime $k^{NO_3}$ values observed during Type-2 nights compared Type-1 nights is accompanied by lower (factor ~2.5) monoterpene mixing ratios (Fig. 3c). The median temperature during Type-2 nights are only up to 1 K colder than compared to Type-1 nights (Fig. 3d), which, based on the expression ($E_{MT} \propto \exp(\beta(T - 297K))$ with β = 0.1 K$^{-1}$, (Guenther et al., 1993)) results in a change of only 10% and is thus insufficient to explain the differences observed in ΣMT on these

nights.

With values of 85-95 %, the median relative humidity (Fig. 3e) was higher by around 5 % (and increased continuously) during Type-1 nights, than for Type 2, for which a much smaller increase from 82 to 87 % was observed.

In summary, in addition to very low NO$_3$ reactivity, Type-2 nights are characterized by (1) larger and constant O$_3$ mixing ratios, (2) lower but constant RH, and (3) low concentrations of reactive trace gases like NO and monoterpenes. These

observations support the presence of a very shallow surface layer with its top located below the tip of the inlet and decoupling of the sampled air from ground-level emissions (i.e. of NO and VOCs). Previous observations of strong gradients in NO$_3$ mixing ratios and low reactivities have showed that decoupling of the air-mass from ground-level emissions can lead to NO$_3$ lifetimes of up to hours (Allan et al., 2002; Brown et al., 2016; Sobanski et al., 2016). In order to test the hypothesis that low



NO₃ reactivities observed during Type-2 nights are the result of sampling from the nocturnal boundary layer (NBL), we
mounted temperature and relative humidity sensors on a multi-copter drone to measure gradients in these parameters on the
night of 22-23rd July, which is the same night as depicted in Fig. 2b.

The drone was located ~ 20 m to the NE of the inlet, the starting height (ground level) was about 12 m lower than the top of
the inlet. The drone flew a vertical profile with the first ascent/descent started before sunset at 18:30 UTC (blue dotted line,
F1 in Fig. 2b) and a second after sunset at 20:20 UTC (red dotted line, F2 in Fig 2b). The flights were restricted to heights of
~ 100 m above ground level owing to operational restrictions in the vicinity of Frankfurt airport.

The gradients in potential temperature θ, for the two flights are shown in Fig. 4a. At 18:30 UTC (blue curve), the potential
temperature increases gradually with altitude (positive stratification) as expected for a well-mixed boundary layer (Stull, 1988;
Brown et al., 2007b). In contrast, the potential temperature gradient measured at 20:20 UTC reveals a strong increase in the
first 3 m, which represents the nocturnal surface layer. Above this, the potential temperature increases more slowly until ca.
20 m above the ground. This zone (shaded in red) represents the stable NBL above which the potential temperature is almost
independent of height (neutral stratification), which is the typical behaviour of the residual layer (Stull, 1988; Brown et al.,
2007b). The gradient in relative humidity (Fig. 4b) after sunset indicates a similar vertical structure with the top of the NBL
characterized by a minimum in the relative humidity (Brown 2007b), also explaining why RH was, on average, lower during
Type-2 compared to Type-1 nights (Fig. 3e). The approximate height of our inlet was situated ca. 10 m above the ground and
the profile of θ implies that the air we sampled was from a NBL decoupled from ground-level emissions and in which vertical
mixing is weak (Brown and Stutz, 2012). Under this scenario, NO originating from soil emissions and VOCs from plant
emissions are trapped in the surface layer and only inefficiently entrained into the NBL. Unfortunately, owing to delays in
obtaining permission to fly the drone, unfavourable weather conditions and other logistical considerations, these two flights
on this one night are the only ones in which vertical profiles of temperature and RH were obtained. None-the-less, these
observations provide important clues to how the meteorological situation can influence NO₃ reactivity and NO levels at inlet
height.

### 3.3 Contribution of VOCs to $k^{NO_3}$

As described above, $k^{NO_3}$ includes the contribution of VOCs only and it is thus expected to correlate with the summed first-
order loss rates, $\Sigma k_i [VOC]_i$ derived from the concentration $[VOC]_i$ of each VOC and the corresponding rate constant ($k_i$) for
its reaction with NO₃, provided that all VOCs with a significant contribution were measured.

Unsaturated organic compounds (often of biogenic origin such as isoprene or terpenes) are generally the dominant reaction
partners for NO₃ in forested environments (Ng et al., 2017). During TO2021, several hundreds of pptv of isoprene,
monoterpenes and sesquiterpenes were detected during the second half of the campaign when VOC measurements became
available (see Fig.1 and Fig.S4). Owing to their low rate coefficients (IUPAC, 2022), alkanes, aromatics and saturated,
oxygenated species such as acetaldehyde, acetone and methanol were found to contribute negligibly to $k^{NO_3}$. Consequently,





only isoprene and the sum of mono- and sesquiterpenes are relevant for analysis. GC-MS measurements from a previous summer campaign at this site (Sobanski et al., 2017) derived fractional contributions to ΣMT of 50.5%, 28.9% and 20.6% for α-pinene, limonene and myrcene, respectively. Using an accordingly weighted average of evaluated kinetic data (IUPAC, 2022), we derived an effective rate constant of $k = 8.9 \times 10^{-12}$ cm$^3$ molecule$^{-1}$ s$^{-1}$ for NO$_3$ + monoterpenes reactions at this site.

To calculate NO$_3$ loss rates resulting from its reaction with sesquiterpenes, we used the IUPAC-recommended rate coefficient for NO$_3$ + β-caryophyllene. Neglecting the uncertainty associated with the assumption that the MT mixture was the same in both campaigns and combining the uncertainty in the measured VOC mixing ratios (20 %) and in the effective rate coefficient (25 %) leads to an overall fractional uncertainty of 33 % in each term of $\Sigma k_i[VOC]_i$.

In Fig. 5a we present a time-series of $k^{NO_3}$ and k$_i$[VOC]$_i$. Clearly, $k^{NO_3}$ and $\Sigma k_i[VOC]_i$ agree within associated uncertainties

most of the time. The poorer agreement observed around the 16$^{th}$ July may have been related to the presence of fog and droplets in the sampling line and that around the 24$^{th}$ July was most probably caused by conditioning effects when switching between flow-tubes. As indicated by the area in purple, the NO$_3$ reactivity was almost entirely determined by the reaction with monoterpenes. Figure 5b focusses on the Type-2 night previously shown in Fig. 2b (but all $k^{NO_3}$ < LOD set to 0.006 s$^{-1}$) suspected to be impacted by a boundary layer effect. Within associated uncertainties, the VOC measurements confirm that

VOC-induced NO$_3$ reactivities are close to or below 0.006 s$^{-1}$ for this period. The average contribution of the VOCs to $\Sigma k_i[VOC]_i$ is depicted in Fig. 5c and shows that 84% of the overall reactivity is caused by monoterpenes, while isoprene and sesquiterpenes contribute 7% and 9% respectively.

Figure 6 plots $\Sigma k_i[VOC]_i$ versus $k^{NO_3}$ for which a bivariate regression yields a slope of 1.04 ± 0.03 (2σ) and an intercept of $(6.6 \pm 0.4) \times 10^{-3}$ s$^{-1}$. A slope close to unity suggests near closure for the NO$_3$ reactivity budget while the intercept is the

equivalent to the reactivity caused for example by 27 pptv of β-caryophyllene or an overestimation of NO by just 18 pptv. We recall however, that speciated monoterpenes were not measured in TO2021 and the effective rate constant was based on the (non-testable) assumption that the summertime monoterpene composition at this site has remained unchanged over the last 10 years. The true uncertainty associated with the slope is expected to be close to 30%, suggesting that the very good agreement may be partially fortuitous. None-the-less, we can conclude that the vast majority of the reactivity measured directly results

from NO$_3$ + monoterpene interactions.

### 3.4 Fractional contribution of VOCs to NO$_3$ losses throughout the diel cycle

The dominant, direct gas-phase loss of NO$_3$ occurs via photolysis ($J_{NO_3}$) reaction with NO ($k_7$[NO]) and reaction with VOCs ($k^{NO_3}$). Neglecting depositional losses of NO$_3$, the fractional contribution $F$ of $k^{NO_3}$ to the overall NO$_3$ loss rate constant, $L_{NO_3}$, is thus given by:

$F = \dfrac{k^{NO_3}}{L_{NO_3}} = \dfrac{k^{NO_3}}{k^{NO_3} + J_{NO_3} + k_7[NO]}$                              (Eq.2)





Based on measured $k^{NO_3}$, [NO] and $J_{NO_3}$ (calculated from actinic flux measurements), we calculated time dependent values of each loss process throughout the campaign. The resulting mean diel cycle of *F* is depicted in Fig. 7.

During the daytime, photolysis and reaction with NO were the dominant loss processes for $NO_3$, as expected. The fractional contribution of VOC-induced losses is low at noon (~ 9 %) but increases to up to 30% in the afternoon. The $NO_X$ levels at this

site are such that, between sunrise and sunset, reaction with NO is on average (± 1σ) the dominant loss process for $NO_3$ (53 ± 20 %), followed by photolysis (31 ± 19 %) and reaction with VOCs (16 ± 15 %). This non-negligible contribution of VOCs to the daytime losses of $NO_3$ is in broad agreement with field measurements in a boreal forest in Finland and on top of the Hohenpeissenberg mountain, where values of ~20 % were reported (Liebmann et al., 2018a; Liebmann et al., 2018b). This underlines that $NO_3$, often considered to be important only at night, also contributes to the oxidation of BVOC during the day

and thus potentially to the formation of organic nitrates (in competition to OH- and $O_3$-initiated oxidation) throughout the diel cycle for example (Liebmann et al., 2019; Foulds et al., 2021).

At nighttime, in the absence of actinic radiation (to convert $NO_2$ to NO) and local anthropogenic emissions, NO levels are generally suppressed by reaction with $O_3$. Fig. 7 reveals that 50-60 % of $NO_3$ was lost via reaction with VOCs at nighttime during TO2021, the remaining fraction reacting with NO (R7). The contribution of NO to the nighttime $NO_3$ reactivity is larger

than previously observed with the $k^{NO_3}$-FT-CRDS instrument where reaction with VOCs was identified as the only significant loss process (Liebmann et al., 2018a; Liebmann et al., 2018b). A significant average contribution from NO is readily understood when one considers the large rate coefficient for reaction with $NO_3$ ($k_7$ = 1.8 x $10^{-11}$ cm$^3$ molecule$^{-1}$ s$^{-1}$ at 298 K (IUPAC, 2022)) and NO mixing ratios well above the detection limit on many nights. Fig. 8 reveals a large night-to-night variability in the NO mixing ratio with minimum values close to the detection limit and maxima > 80 pptv. In the absence of

local anthropogenic sources, soil emissions constitute the most likely source of NO at this site. Assuming that reaction with $O_3$ represents the only NO loss process, and that stationary state as in Eq. 3 is achieved (a valid assumption as the lifetime of NO is only a few minutes in the presence of 20-40 ppbv of $O_3$) NO emission rates ($E_{NO}$) of 0.18 to 0.47 pptv s$^{-1}$ are necessary to reproduce the observed nighttime NO mixing ratio within a surface layer of 10 m height.

$$E_{NO} = [NO] \cdot k_1[O_3] \tag{Eq. 3}$$

In the absence of measurements of NO soil emission fluxes at the site and recognising that that these are highly dependent on temperature, season, soil humidity and degree of fertilization (Pilegaard, 2013), we take an annual mean NO emission flux of 1 kg ha$^{-1}$ yr$^{-1}$ for temperate, uncultivated grassland (Ludwig et al., 2001) to derive (assuming the same layer height of 10 m) an NO emission rate of 0.27 pptv s$^{-1}$, which lies within the range quoted above. As the summit of the Kleiner Feldberg is covered with blueberry bushes and surrounded by coniferous forest and that soils impacted from blueberry plants or spruce

can support higher NO net fluxes than grass-covered soils (Bargsten et al., 2010), significant NO soil emissions at the summit of the Kleiner Feldberg appear to be plausible. Figure 8 also reveals that the highest levels of NO observed at 10 m height occur when $O_3$ values are lowest. Anti-correlated NO and $O_3$ mixing ratios are often observed when plumes of freshly emitted NO is mixed into aged air masses containing $O_3$ and is a result of reaction R1 which converts NO to $NO_2$. For our observations at 10 m height, chemistry (temperature dependent kinetics), boundary layer dynamics (extent of mixing/decoupling of surface





layer and NBL) and plant physiology (emission rates of NO) may all contribute to the extent to which NO and $O_3$ react. As the large night-to-night variability in the NO mixing ratios cannot be explained by temperature-dependent changes in the rate coefficient $k_1$ or in the emission rate of NO, we conclude that boundary layer effects dominate and that the height of the surface layer and the degree to which NO is entrained from the surface layer into the NBL are the main controlling factors. We consider two limiting cases: 1) When the top of the nocturnal surface layer is above the inlet, and mixing is sufficient to homogenize

the air within the first 10 m above the ground, NO originating from the soil can react with $O_3$ via R1 (Aneja et al., 2000). This would correspond to observations during Type-1 nights. 2) When the surface-layer is less than 10 m deep and is decoupled from the NBL, soil emitted NO is not sampled by the inlet (ca. 10 m above the ground) and the measured NO mixing ratios are at the instrument's LOD. In this case, levels of $O_3$ in the NBL remain high, as e.g. observed around 21 July. In reality, trace-gas gradients within the lowest layers will control the extent of mixing and case 1) will only operate when high wind

speeds induce turbulent mixing close to the surface. We conclude that the variability in nighttime NO and the anti-correlation with $O_3$ (see Fig. S5a) reflect rapid changes in boundary layer dynamics and vertical mixing within the lowest layers. Similarly high variability in $NO_3$ mixing ratios has been attributed to a related phenomenon (Crowley et al., 2011). We note that if the time-scales over which boundary-layer dynamic change is less than the lifetime of NO, our steady-state assumption breaks down. None-the-less, the presence of up to 90 pptv of NO at nighttime in the presence of 20-40 ppbv of $O_3$ implies significant

production of $NO_2$.

We examined the nighttime generation of $NO_2$ using box model calculations (FACSIMILE/CHEMCAT (Curtis and Sweetenham, 1987)) employing Reactions R1 to R4 and R7 with IUPAC-recommended, temperature-dependent rate coefficients (S5 of the Supplement) and constrained by measurements of NO, $O_3$, ambient temperature and pressure. Known loss processes for $NO_2$ at night are the slow reaction with $O_3$ (to form $NO_3$) and with $NO_3$ to form $N_2O_5$ (R2–R4) and deposition

to surfaces (e.g. foils, soil). Note that this simulation considers R1 as the only $NO_2$ source and that it is only valid if chemistry and transport happen on a similar time scale.

Figure 9 plots the measured nighttime $NO_2$ mixing ratios (black symbols) together with the model output using $vd_{NO2} = 0.015$ cm s$^{-1}$ (which is based on a mean nighttime $NO_2$ deposition for foliar surfaces (Delaria et al., 2018) and a value that is a factor 10 larger ($vd_{NO2} = 0.15$ cm s$^{-1}$) in both cases assuming a surface layer height of 10 m to derive loss rate constants of $1.5 \times 10^{-5}$

and $1.5 \times 10^{-4}$ s$^{-1}$ respectively. Clearly, the larger deposition velocity is necessary to roughly align measured and simulated $NO_2$ mixing ratios. Such large $NO_2$ deposition velocities have previously been evoked in order to bring observed $NO_2$ levels and NO emission rates into agreement (Jacob and Wofsy, 1990) and our average, nighttime deposition velocity of 0.15 cm s$^{-1}$ is comparable to values of 0.1-0.57 cm s$^{-1}$ determined in boreal coniferous forests (Rondon et al., 1993) at night and 0.096 cm s$^{-1}$ obtained in a temperate coniferous forest (Breuninger et al., 2013).

The interaction of $NO_2$ with foliar surfaces, which can serve as both source and sink of $NO_2$ is complex (Breuninger et al., 2013; Delaria et al., 2018) and a scenario in which the high (but variable) nighttime NO mixing ratios result from soil emissions while $NO_2$ is simultaneously deposited on foliar surfaces is conceivable. Given that the stratification of the lowermost atmosphere at TO2021 was only examined on one night, and considering the likely variability in NO emission rates and $NO_2$



deposition velocities (Ludwig et al., 2001; Ganzeveld et al., 2002), our interpretation of the nighttime NO and $NO_2$ data remains
speculative. Considering the lack of correlation between wind direction and abundance of nighttime NO (Fig. S5b), an
alternative, point NO emission source (e.g. an NO bottle, or exhaust line) seems unlikely. Interferences by other trace-gases
or reasons for bias of the CLD instrument could not be identified as causes for the high nocturnal levels of NO.

## 3.5 NO₃ mixing ratios

During the TO2021 intensive, ambient $NO_3$ mixing ratios were not monitored. However, as both the total loss term $L_{NO_3}$ and
the production term $(P_{NO_3} = (k_2[NO_2][O_3]))$ are known, we can derive $NO_3$ mixing ratios by assuming that $NO_3$ is in steady-
state, i.e. that loss and production are balanced and the derivative of the $NO_3$ mixing ratios is independent of time. Steady-state
calculations of $NO_3$ lifetimes or $NO_3$ mixing ratios have been carried out in numerous studies (Platt et al., 1984; Geyer and
Platt, 2002; Brown et al., 2011; Crowley et al., 2011; Liebmann et al., 2018a; Liebmann et al., 2018b) and have shown to be
valid, when $NO_3$ reactivities are high enough and the chemical equilibrium to $N_2O_5$ (R3 and R4) is not perturbed by sudden
changes in $NO_2$ mixing ratios (Brown et al., 2003a; Dewald et al., 2020). Steady-state $NO_3$ mixing ratios can be calculated
with Eq. 4,

$$[NO_3]_{ss} = \frac{P_{NO_3}}{L_{NO_3}} = \frac{k_2[NO_2][O_3]}{k^{NO_3} + J_{NO_3} + k_7[NO]} \tag{Eq. 4}$$

which neglects both direct and indirect, heterogeneous loss of $NO_3$ (R8 and R9). Previous estimates of the $NO_3$ loss by aerosol
uptake on the Kleiner Feldberg returned values of $\approx 0.001$ s$^{-1}$ or lower (Crowley et al., 2010; Phillips et al., 2016; Sobanski et
al., 2016) and are consequently insignificant compared to the average nighttime overall $NO_3$ loss rate of $\approx 0.03$ s$^{-1}$.
Figure 10 displays a time-series of the calculated overal $NO_3$ loss-constant, production rate and steady-state mixing ratios for
TO2021. Nighttime $NO_3$ losses vary typically between $< 0.006$ s$^{-1}$ and 0.03 s$^{-1}$, while the daytime losses were as large as 0.3
s$^{-1}$. The $NO_3$ production rate was, on average, close to $\sim 0.02$ s$^{-1}$ at nighttime, increasing to 0.1 s$^{-1}$ during the day when $NO_2$
and/or $O_3$ mixing ratios were large. $NO_3$ mixing ratios thus calculated are lower than about 6 pptv for all nights (one exception
of 12 pptv on the 10th July) and well below 2 pptv for most of the nights.

### 3.5.1 Comparison with previous NO₃ measurements at the Kleiner Feldberg

$NO_3$ measurements with which to compare the present data-set have been recorded at the Kleiner Feldberg during campaigns
in 2008, 2011, 2012 and 2015 for which key details (including names and acronyms) are summarized in Tab.1:



**Table 1:** Nighttime NO$_3$ mixing ratios, median production rates and median nighttime loss rates at the top of the Kleiner Feldberg.

| Campaign | Reference | Period | # Nights (< LOD) | $P_{NO_3}$ pptv s$^{-1}$ | $L_{NO_3}$ 10$^{-3}$ s$^{-1}$ | NO$_3$ pptv | k$_8$ 10$^{-3}$ s$^{-1}$ | k$_9$ 10$^{-3}$ s$^{-1}$ |
|---|---|---|---|---|---|---|---|---|
| **TO2008** | Crowley et al, 2010 | May 2008 | 6 (0) | 0.033 | 2.2[b] | < LOD – 65[a] | 1.6 | < 0.2 |
| **PARADE** | Sobanski et al., 2016 | Aug-Sep 2011 | 21 (4) | 0.044 | 4.5[b] | < LOD – 250[a] | | 2 |
| **INUIT** | This work | Aug 2012 | 16 (4) | 0.049 | 3.7[b] | < LOD – 190[a] | | |
| **NOTOMO** | Sobanski et al., 2017 | Jul 2015 | 24 (10) | 0.049 | 7.5[b] < 5 – 40[d] | < LOD – 50[a] | | |
| **TO2021** | This work | Jul 2021 | 34 (14) | 0.025 | 27[a] < 6 – 40[c] | 0-12[b] | | |

Notes: # Nights = Number of nights with measurements, the number in brackets represents the number of nights where either the NO$_3$ mixing ratio or the directly measured value of $k^{NO_3}$ was below the LOD. Direct (k$_8$) and indirect (k$_9$) loss rates of NO$_3$ by heterogeneous uptake of NO$_3$ and N$_2$O$_5$ were calculated only for TO2008 and PARADE. [a]directly measured. [b]steady-state calculation. [c]directly measured; VOC contribution only. [d]directly measured for 3 nights, no NO measurements available (Liebmann et al., 2017). TO2008 = Mini (un-named) 
campaign with only NO$_3$, NO, O$_3$ and NO$_2$ measurements. PARADE = PArticles and RAdicals: Diel observations of the impact of urban and biogenic Emissions, INUIT = Ice NUclei research UnIT, NOTOMO = NOcturnal chemistry at the Taunus Observatory: insights into Mechanisms and Oxidation.

The first measurements of NO$_3$ (and N$_2$O$_5$) at the Kleiner Feldberg were performed on 6 nights in May 2008 (Crowley et al.,
2010) (this data set is referred as TO2008), on 21 nights in July 2011 (PARADE campaign (Sobanski et al., 2016)), on 16 nights in August 2012 (INUIT campaign) and in September 2015 during the NOTOMO campaign (Liebmann et al., 2017; Sobanski et al., 2017). All previous NO$_3$ data sets except for INUIT have been published. The time-series of the NO$_3$, NO$_2$, O$_3$ mixing ratios (and resulting $P_{NO_3}$ and $L_{NO_3}$ according to Eq. 4) from each of the campaigns used for this analysis are reproduced in the Supplement (S6, Fig. S6-S9) together with key features of the instruments used (Tab. S1).
The presence of nearby industrial centres imparts a strong wind-direction dependence on the composition of the air (and especially NO$_X$) at the Kleiner Feldberg with densely populated cities (and thus anthropogenic sources of NO$_x$) located in the SE and SW sectors. An overview of the prevailing wind directions and NO$_2$ mixing ratios during each campaign are summarized in Fig. 11. The lowest, average NO$_2$ mixing ratios were encountered during TO2008 (air arriving mainly from the East) and TO2021 which had a large contribution of air mases arriving from the North and West but almost none from the
Frankfurt area (SE-SSE). TO2021 is the only campaign with a significant contribution of air masses arriving from the "clean" Northern sector and the generally lower NO$_X$ levels during TO2021 may also have been a result of changes in vehicle usage in the region as a higher fraction of locally employed people worked from home as a result of the COVID-19 pandemic (Reifenberg et al., 2021).





Figure 12 indicates that, in comparison to the previous summer campaigns (PARADE, INUIT and NOTOMO) the temperatures
were lower during TO2021 with the maximum value of 22 °C being ~10 °C lower than the maximum value during NOTOMO.
TO2021 and PARADE had the highest incidence of very humid days, with a median RH of > 80 % for TO2021 and > 75%
for PARADE while for TO2008 the median relative humidity (~51 %) was the lowest.

For comparison of the nighttime $NO_3$ mixing ratios, periods of daytime-nighttime transitions (when $NO_3$ mixing ratios strongly
change at sunrise or sunset) were excluded. The $NO_3$ mixing ratios (lower panel), loss rates (middle panel) and production
rates (upper panel) for each campaign are depicted in Fig.13 as a box-and-whisker plot. Note that nights on which the $NO_3$
mixing ratios were > 0 but below the instrument's LOD, were taken into account, whereas for the calculation of $L_{NO_3}$ in the
campaigns prior to TO2021, reactivities derived from $NO_3$ mixing ratios below the LOD (i.e. < 1.5 pptv) were excluded from
the analysis so that $L_{NO_3}$ is not biased by values associated with high uncertainties.

Figure 13a shows that, during PARADE, INUIT and NOTOMO, the nighttime $NO_3$ production rates were similar in terms of
both median values (~ 0.05 pptv s$^{-1}$) and range. Throughout these three campaigns, high production rates (above 0.3 pptv s$^{-1}$)
were occasionally observed, which for PARADE (Sobanski et al., 2016) were linked to winds originating from urban regions.
Figure 13a also reveals that the median, nighttime $NO_3$ production rates during PARADE, INUIT and NOTOMO were higher
than during TO2008 and TO2021 (0.033 and 0.025 pptv s$^{-1}$) which was driven by the lower $NO_2$ mixing ratios in TO2008 and
TO2021 for which air from the cleaner easterly and northerly sectors was encountered more frequently. Campaign-averaged
diel cycles of $O_3$ in the Supplement (Fig. S10) indicate that $O_3$ during TO2008 and TO2021 were not substantially lower (even
higher in the case of TO2008) than during PARADE, INUIT and NOTOMO.

Figure 13b shows clearly that, with a median value of 0.028 s$^{-1}$, the nighttime $NO_3$ loss rates ($L_{NO_3}$) during TO2021 were
significantly higher than for all other campaigns, which were 0.0075 s$^{-1}$ for NOTOMO, 0.0045 s$^{-1}$ for  PARADE, 0.0037 s$^{-1}$
for INUIT and 0.0022 s$^{-1}$ for TO2008. A partial explanation for the greater $NO_3$ loss term during TO2021 is found in the
nighttime NO mixing ratios, which were significantly larger than those measured in e.g. TO2008 or PARADE. The effect of
removing the contribution of NO reaction to $L_{NO_3}$ during PARADE (and TO2008) is minimal, as NO was close or below the
LOD (4-10 pptv) on most nights (Crowley et al., 2010; Sobanski et al., 2016), which is confirmed by the corresponding
campaign-averaged diel cycles of NO mixing ratios (Fig. S11). In contrast, subtraction of the contribution to $NO_3$ reactivity of
the high nighttime levels of NO observed during TO2021, would reduce $L_{NO_3}$ to ~0.011 s$^{-1}$ (red, horizontal line in Fig. 13b)
which is more comparable to that observed during e.g. NOTOMO and PARADE.

We note that, in general, the comparison of $NO_3$ loss rates derived via the steady-state method and direct reactivity may be
complicated by the fact that the steady-state method only works when $NO_3$ is above the detection limit (often a result of low
reactivity) whereas the direct measurement of $NO_3$ losses performs best when reactivities are high. However, $\Sigma k_i[VOC]_i$
suggests that $L_{NO_3}$ was never below 0.002 s$^{-1}$ on Type-2 nights. As shown in the Supplement, setting values of $k^{NO_3} < 0.006$
s$^{-1}$ to 0.002 s$^{-1}$  would only occasionally lead to $[NO_3]_{ss}$ > 10 pptv (Fig. S12a) and thus only have a small impact on the
distribution of $NO_3$ mixing ratios (Fig. S12b), so that this bias cannot be fully responsible for the observed difference.





None-the-less, Fig. 13a and 13b indicate that TO2021 was exceptional in that $P_{NO_3}$ was the lowest of all campaigns at the Kleiner Feldberg while $L_{NO_3}$ was the highest, which result in a calculated median $NO_3$ mixing ratio of just 0.7 pptv. This contrasts greatly with median $NO_3$ mixing ratios of 15, 10, 11 and 4 pptv observed during TO2008, PARADE, INUIT and
NOTOMO (Fig. 13c) on the Kleiner Feldberg.

As alluded to above, this difference is partially caused by unusually high nighttime NO levels, but also results from the low $NO_3$ production rate during TO2021. During PARADE, INUIT and NOTOMO, $NO_3$ mixing ratios above 100 pptv were measured and linked to nights with exceptionally long $NO_3$ lifetimes. For PARADE, this was suggested to be a result of sampling from above the surface layer, where $NO_3$ lifetimes can be large owing to the decoupling from ground-level emissions
(Brown et al., 2003b; Sobanski et al., 2016). While there is evidence for a similar situation for TO2021 on 21 Jul (Fig. 4), in the absence of vertically resolved meteorological data on the other nights, it is not clear whether purely meteorological effects are responsible for the observed low reactivities on 14 nights or whether reduced emission rates of reactive trace-gases additionally play a role. We are presently developing a drone-borne $NO_3$ instrument to provide vertical gradients in $NO_3$ (as well as T and RH) in order to help resolve this issue.

## 4 Summary and conclusions

The fate of the $NO_3$ radical at the summit of the Kleiner Feldberg during the TO2021 intensive in July 2021 was assessed with the help of direct $NO_3$ reactivity and VOC measurements. Directly measured $NO_3$ reactivities towards VOCs ($k^{NO_3}$) were on average ~ 0.011 s$^{-1}$ at night and as large as ~ 0.04 s$^{-1}$ during the day. $NO_3$ reactivities derived from VOC measurements showed an excellent agreement with $k^{NO_3}$ throughout the diel cycle with VOC-induced $NO_3$ losses by monoterpenes dominating with
a contribution of > 80 %. Sesquiterpenes and isoprene contributed with 9 ± 5 % and 7 ± 4 %, respectively.

During the daytime, NO removed on average 53 ± 20 % of the $NO_3$, photolysis and reaction with VOCs contributed ~31 ± 19 and ~ 16 ± 15 % respectively. The daytime contribution of VOC-induced reactivity was highly variable and ranged from ca. 10 % at noon to 30 % in the afternoon implying that $NO_3$ can contribute significantly e.g. to alkyl nitrate formation during daytime.

$k^{NO_3}$ was predominantly below the LOD of 0.006 s$^{-1}$ on 14 of the 34 nights. On one night, for which a vertical temperature and RH gradient were measured, the low $NO_3$ reactivity was associated with reduced vertical mixing and the decoupling of a shallow surface layer from the layer above in which the trace-gas inlet was situated.

In the absence of direct measurements, $NO_3$ mixing ratios during TO2021 were calculated from the total loss rate constant (VOCs, photolysis, NO) and the $NO_3$ production rate to enable comparison with directly measured $NO_3$ mixing ratios during
four previous campaigns between 2008 and 2015 at the Kleiner Feldberg. For TO2021, $NO_3$ loss rates were ca. a factor 3-5 higher than during previous campaigns while $NO_3$ production rates were the lowest. Consequently, the calculated steady-state mixing ratios of $NO_3$ are much lower than those directly measured during TO2008, PARADE and INUIT and NOTOMO. The exceptionally high nighttime $NO_3$ loss rates during TO2021 are partially related to the presence of several tens of pptvs of NO,



so that VOC-induced losses were 50-60 % of the overall loss term. This is in stark contrast to previous observations in forested

environments where reactions with VOCs were the only relevant nighttime loss path of $NO_3$. The observation of NO at levels of 20-80 pptv at nighttime in the presence of 30-40 ppbv of $O_3$ imply large rates of $NO_2$ formation. Constrained box-model calculations suggest that rapid losses of $NO_2$ via e.g deposition would necessary in order to reproduce the observed nighttime $NO_2$ mixing ratios. In order to confirm this hypothesis, measurements of NO emission and $NO_2$ deposition rates on the Kleiner Feldberg under similar meteorological conditions are necessary.

Overall, the intercomparison of the $NO_3$ mixing ratios and $NO_3$ reactivity revealed high variability in data obtained over a long period on the same site and emphasizes that not only chemical effects but also boundary-layer dynamics and plant-physiological processes may have a great impact on observations.

*Coda and Data Availability.* Data of the TO2021 campaign is available upon request at https://keeper.mpdl.mpg.de/ to all

scientists agreeing to the data protocol. The data of all other campaigns is available upon request from the corresponding author. The FACSIMILE code used for the box model can be found in the Supplement (S5).

*Author contributions.* PD measured $NO_3$ reactivity and $NO_2$ mixing ratios during TO2021, analysed the data and wrote the manuscript. JNC organized the TO2021 campaign, measured $NO_x$ and helped to revise the manuscript. CMN and HF provided

NO and $NO_2$ data. AR, AE and JW provided VOC data. JNC and CMN provided $O_3$ data. JS measured actinic fluxes and performed vertical profile measurements of temperature, relative humidity and pressure with the drone. All authors commented on the manuscript.

*Competing interests.* The authors declare that they have no conflict of interest.


*Acknowledgements.*

We thank Andreas Kürten and Joachim Curtius (Institute for Atmospheric and Environmental Sciences, Goethe University Frankfurt am Main) for logistical support and access to the facilities at the Taunus Observatory. We thank the DWD for the provision of meteorological data and Chemours for the FEP sample used to coat the flowtube reactors.

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



# Figures

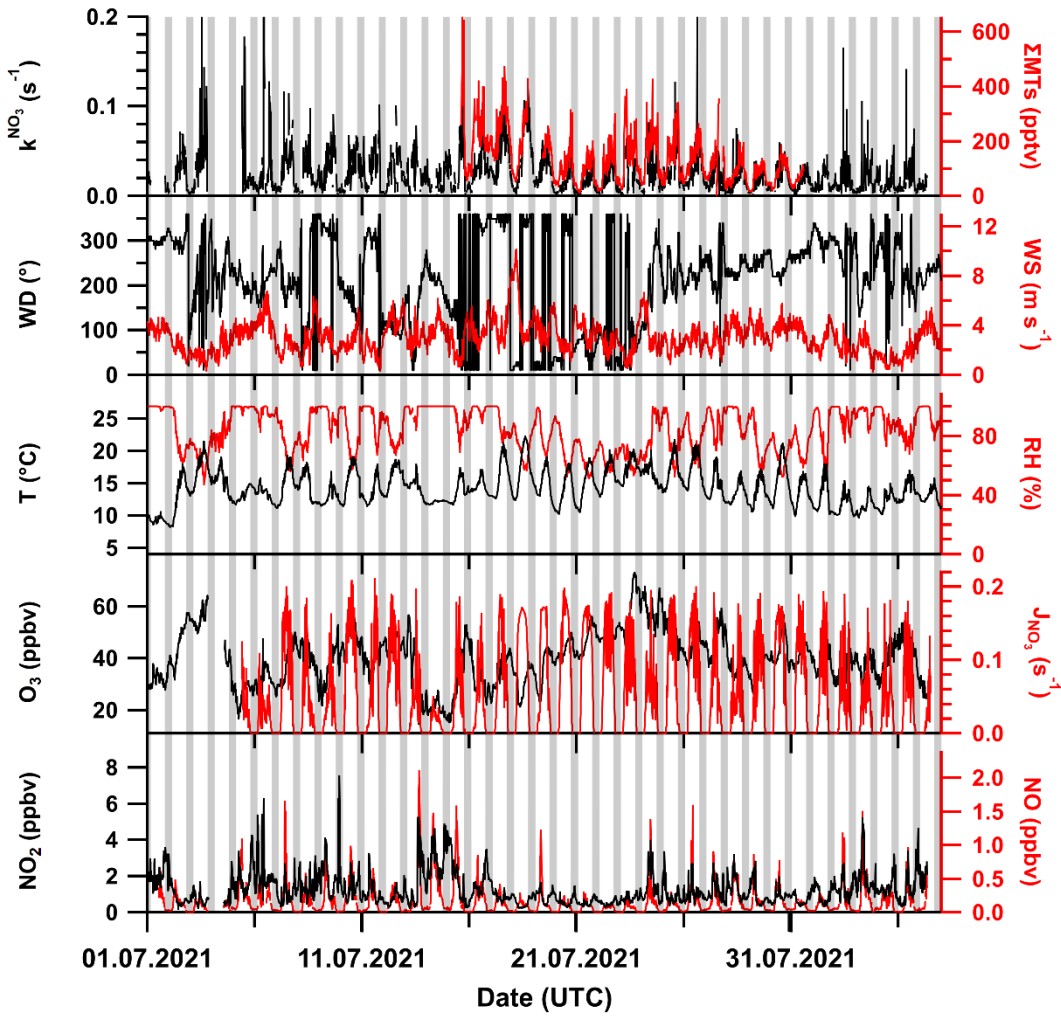

**Figure 1:** Overview of key measurements during the TO2021 campaign with wind direction (WD), temperature (T), sum of monoterpenes (ΣMT), wind speed (WS), relative humidity (RH), NO$_3$ photolysis rate coefficient ($J_{NO_3}$). Meteorological data was provided by the German Meteorological Service (DWD). Nighttime periods are shaded grey. The x-axis ticks are at 00:00 UTC.





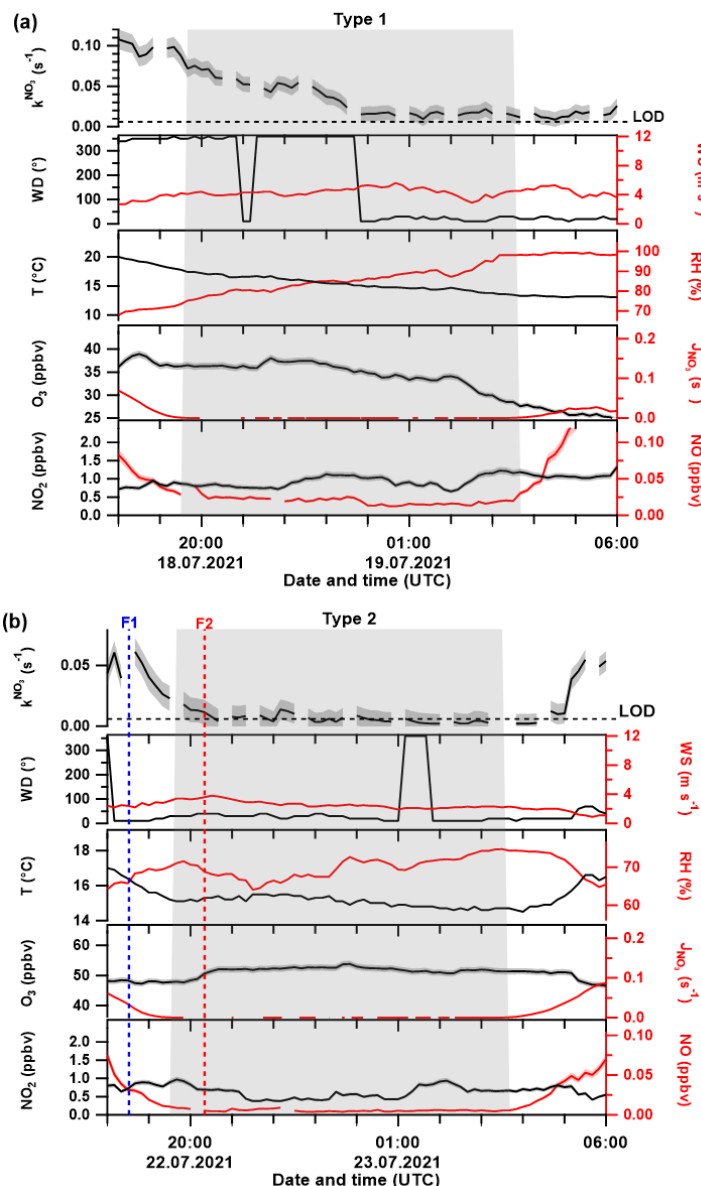

**Figure 2:** Time-series of directly measured NO₃ reactivity ($k^{NO_3}$) together with auxiliary measurements during Type-1 (a) and Type-2 nights night (b). F1 and F2 mark times at which drone-assisted temperature and relative humidity profiles wer measured. The grey-shaded area represents nighttime. Abbreviations are defined in caption of Fig.1. The shaded areas in the colour of the lines denotes the corresponding uncertainty of the measured parameter.




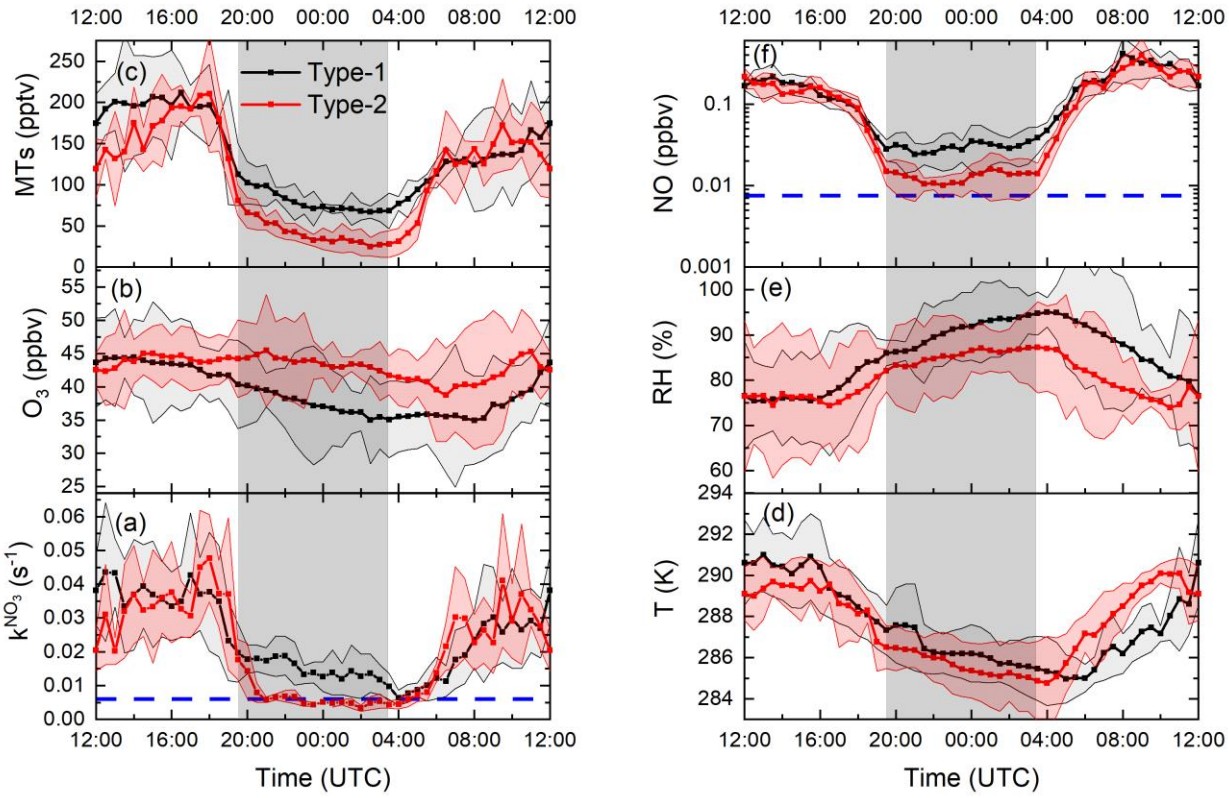

**Figure 3:** Median diel profiles of (a) directly measured $NO_3$ reactivities, (b) $O_3$ mixing ratios, (c) monoterpenes, (d) temperature, (e) relative humidity, and (f) NO mixing ratios classified by night types (Type-1 in black, Type-2 in red). The grey shaded area represents the nighttime period. The shaded areas in line colour represent the 25th and 75th percentiles. The blue lines denote the LODs of the instruments used to measure $NO_3$ reactivity and NO.





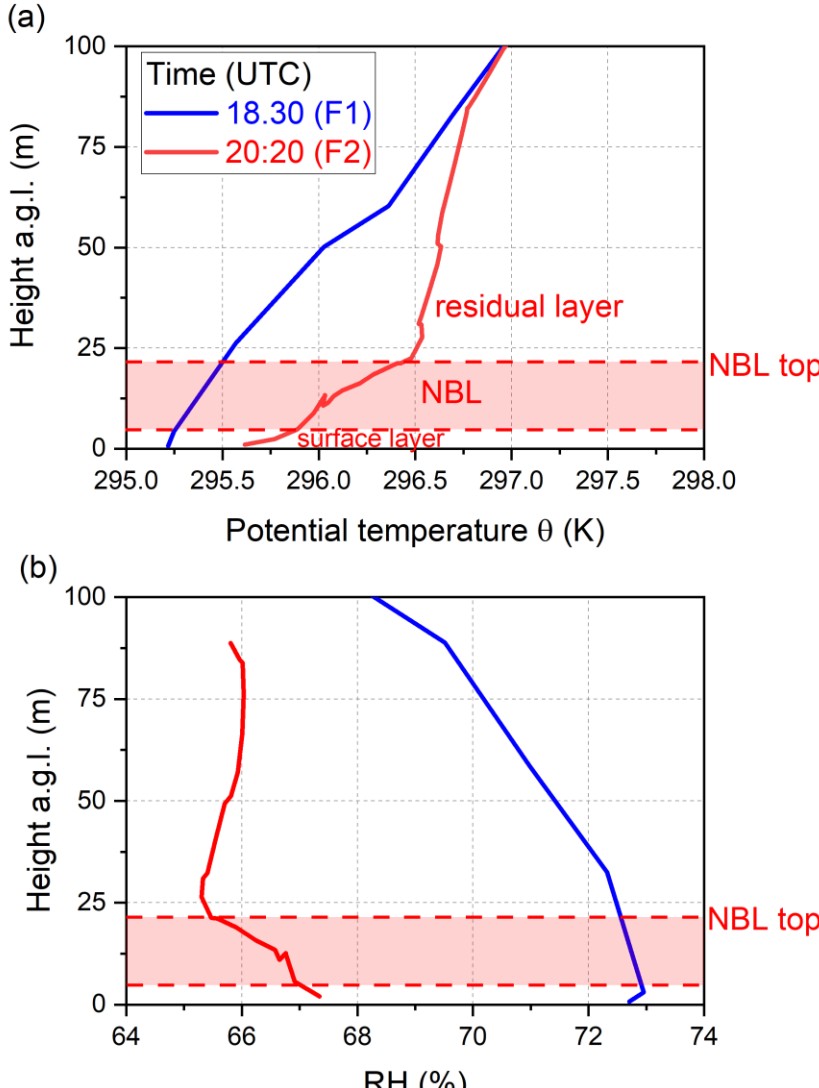

**Figure 4:** Vertical profiles of potential temperature (a) and relative humidity (b) at the summit of the Kleiner Feldberg at 18:30 UTC (blue) and 20:20 UTC (red). The nocturnal boundary layer (NBL) at 20:20 UTC is shaded red.





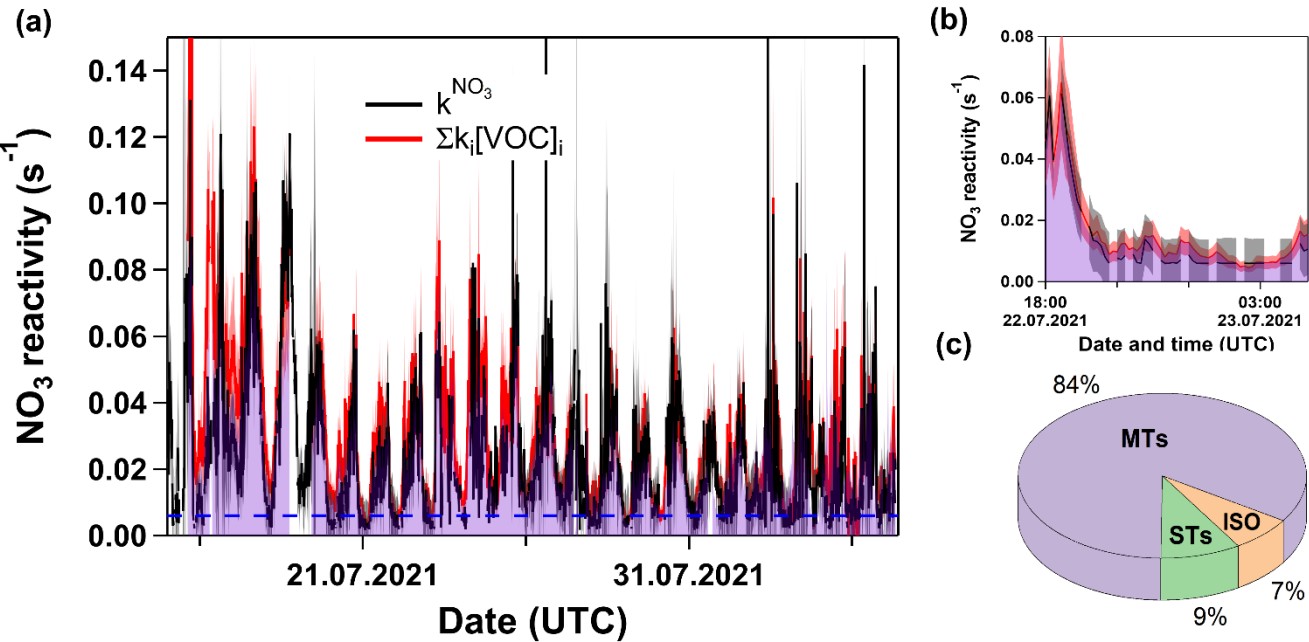

**Figure 5:** (a) Time-series of $k^{NO_3}$ and $\Sigma k_i[VOC]_i$. Dashed blue line marks the LOD of the $k^{NO_3}$ measurement. The purple shade represents the contribution of monoterpenes. (b) Same as (a) but with a detailed view of the night between the 22nd and 23rd July presented in Fig.2b (c) Pie-chart of fractional contributions of isoprene, monoterpenes and sesquiterpenes to $\Sigma k_i[VOC]_i$ over this time period.



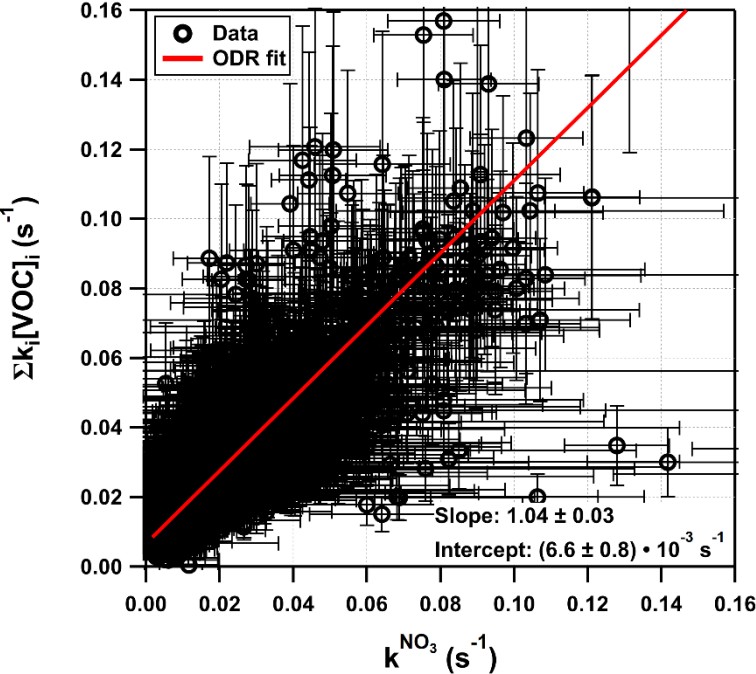

**Figure 6:** Plot of $k^{NO_3}$ versus $\Sigma k_i[VOC]_i$. The red solid line represents an orthogonal distance regression (ODR) with a slope of 1.04 and an intercept of 6.6 x 10$^{-3}$ s$^{-1}$.




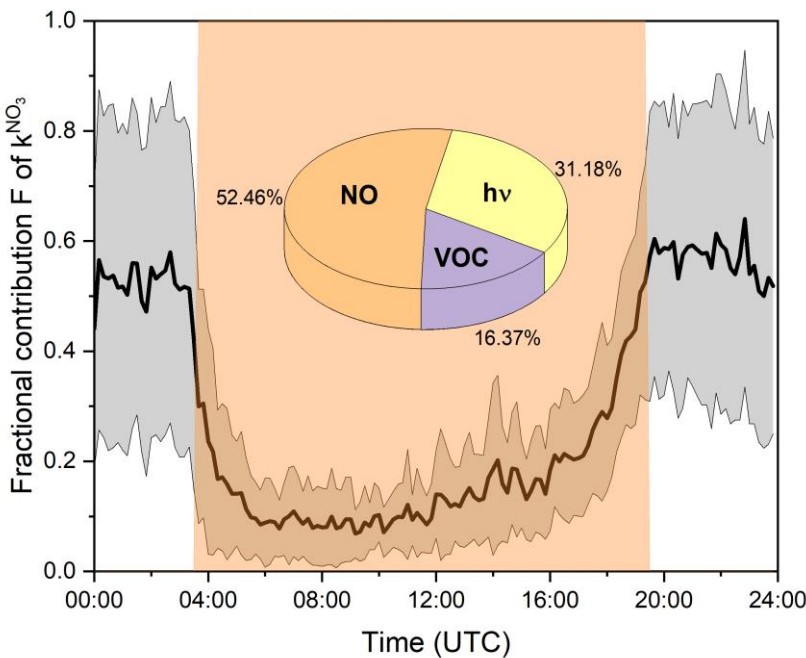

**Figure 7:** Mean, fractional contribution (F) of $k^{NO_3}$ to the overall NO$_3$ loss rate over the diel-cycle. The grey shaded area represents the standard deviation (1σ) of the mean values. Orange shaded area indicates daytime. The pie-chart shows the mean fractional contribution to NO$_3$ loss of reaction with NO, photolysis and reaction with VOCs during the daytime.







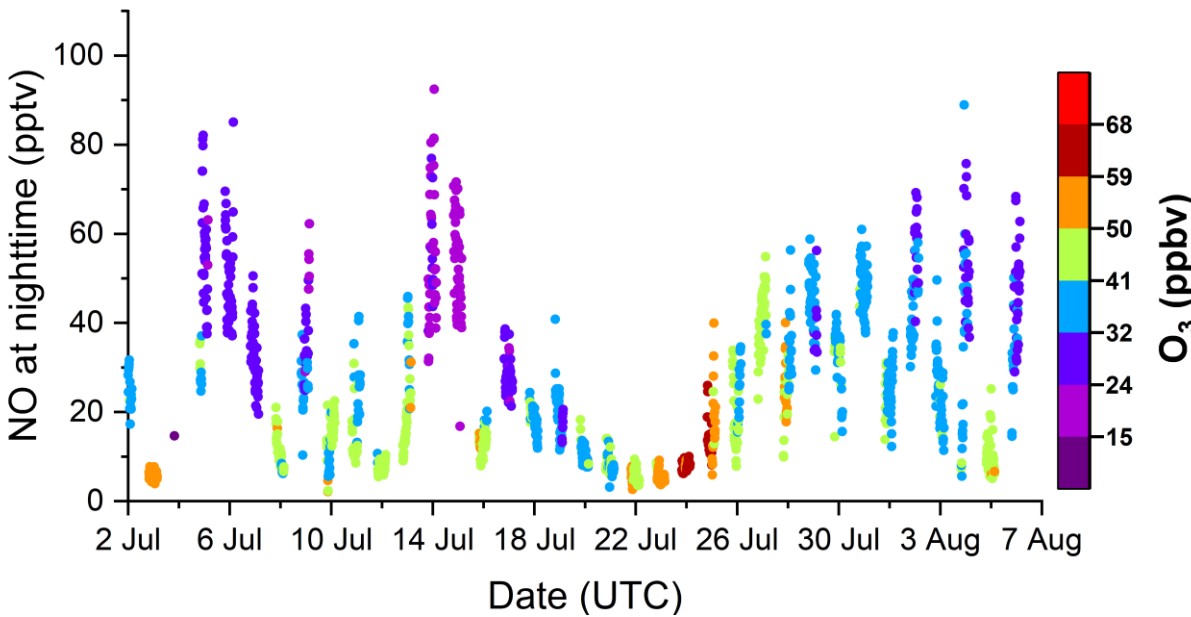

**Figure 8:** Nighttime NO mixing ratios (colour-coded by $O_3$ mixing ratios) during TO2021. The x-axis Ticks represent 00:00 UTC.






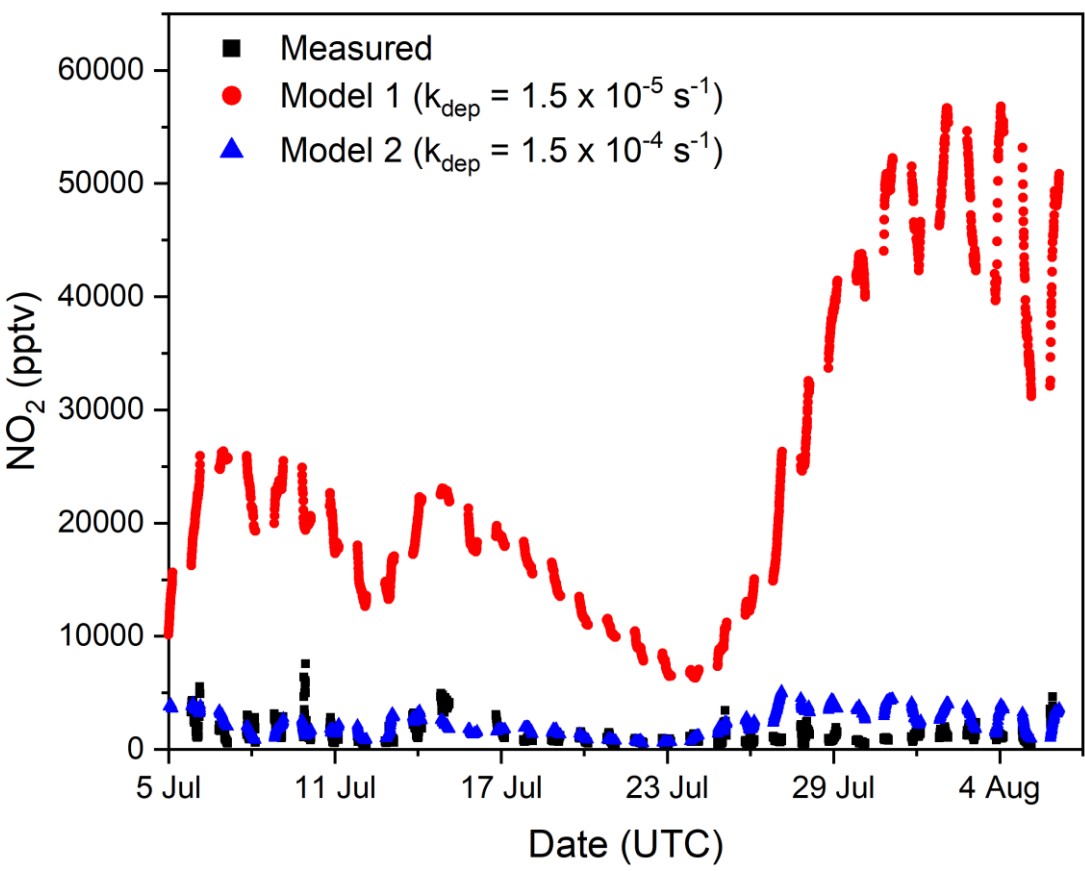

**Figure 9:** Time-series of measured nighttime $NO_2$ mixing ratios during TO2021 (black squares) and modelled $NO_2$ mixing ratios using deposition loss constants of $1.5 \times 10^{-5}$ $s^{-1}$ (Model 1, red circles) and $1.5 \times 10^{-4}$ $s^{-1}$ (Model 2, blue triangles).





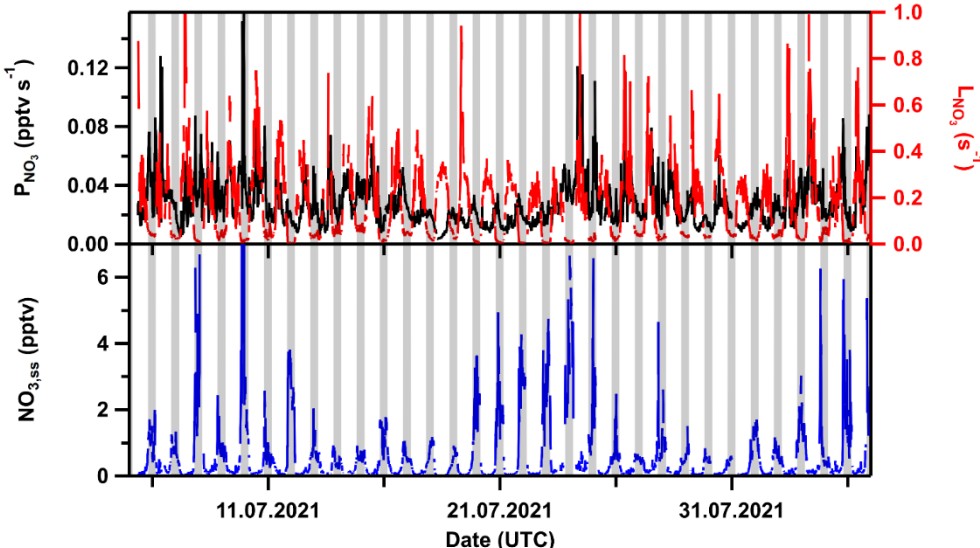

**Figure 10:** Upper panel: NO$_3$ production (left) and loss rates (right) during TO2021. Lower panel: steady-state NO$_3$ mixing ratios. Ticks represent 00:00 UTC. Grey shaded areas denote nighttime.



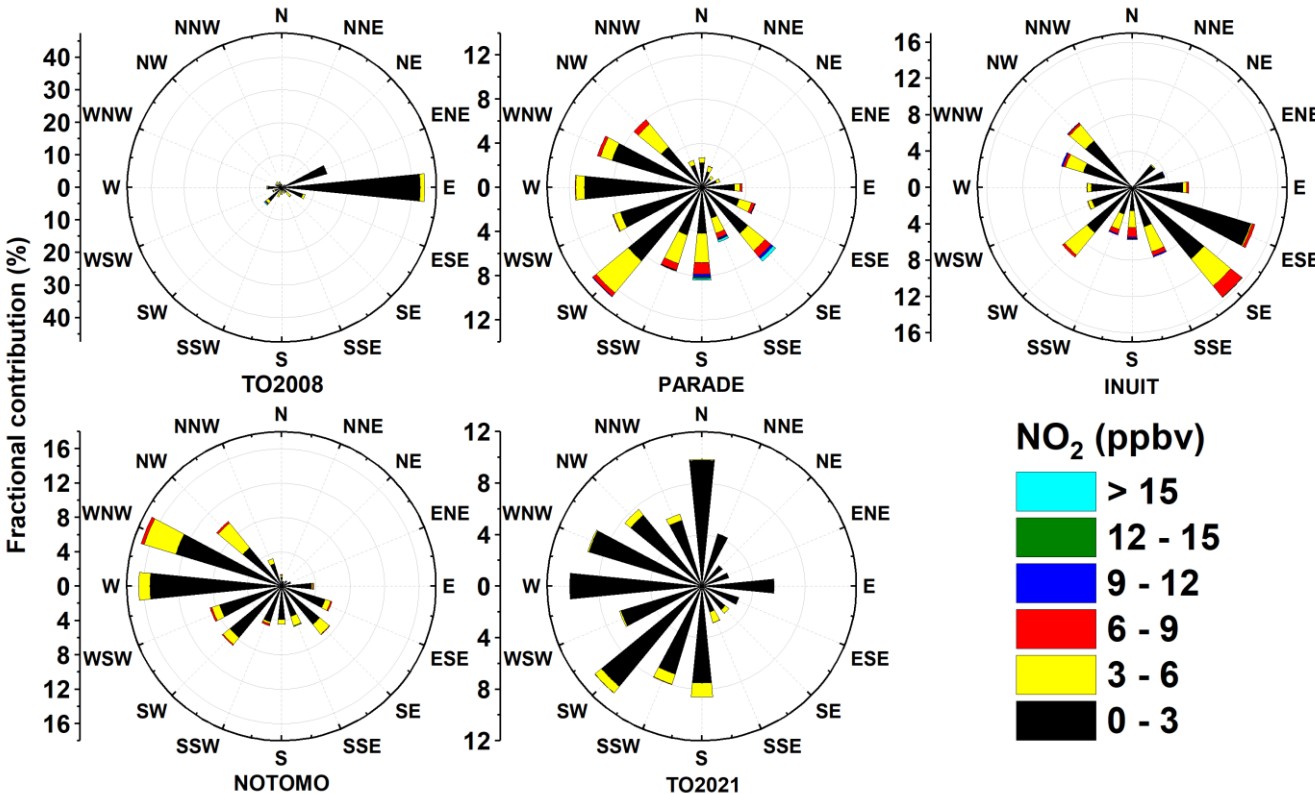

Figure 11: Wind roses indicating the dependence of NO₂ mixing ratio on the wind direction during TO2008, PARADE, INUIT, NOTOMO and TO2021. Wind directions were provided by HLNUG for TO2008 and NOTOMO, by a weather station in PARADE and INUIT (Drewnick et al., 2012) and by the German meteorological service DWD in TO2021.





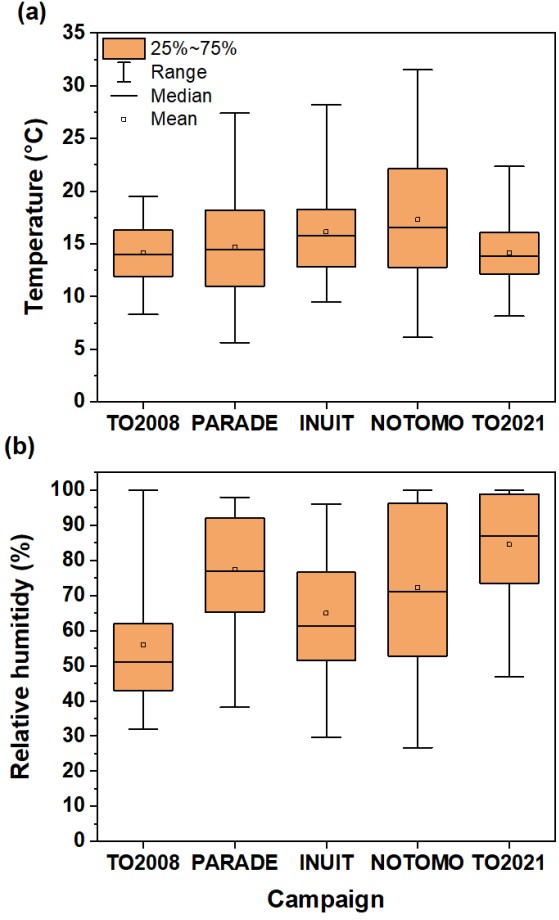

**Figure 12:** Distributions of (a) temperature and (b) relative humidity during five campaigns at the Kleiner Feldberg between 2008 and 2021. Boxes represent the range between the first and third quartiles, whiskers denote the full range of values.



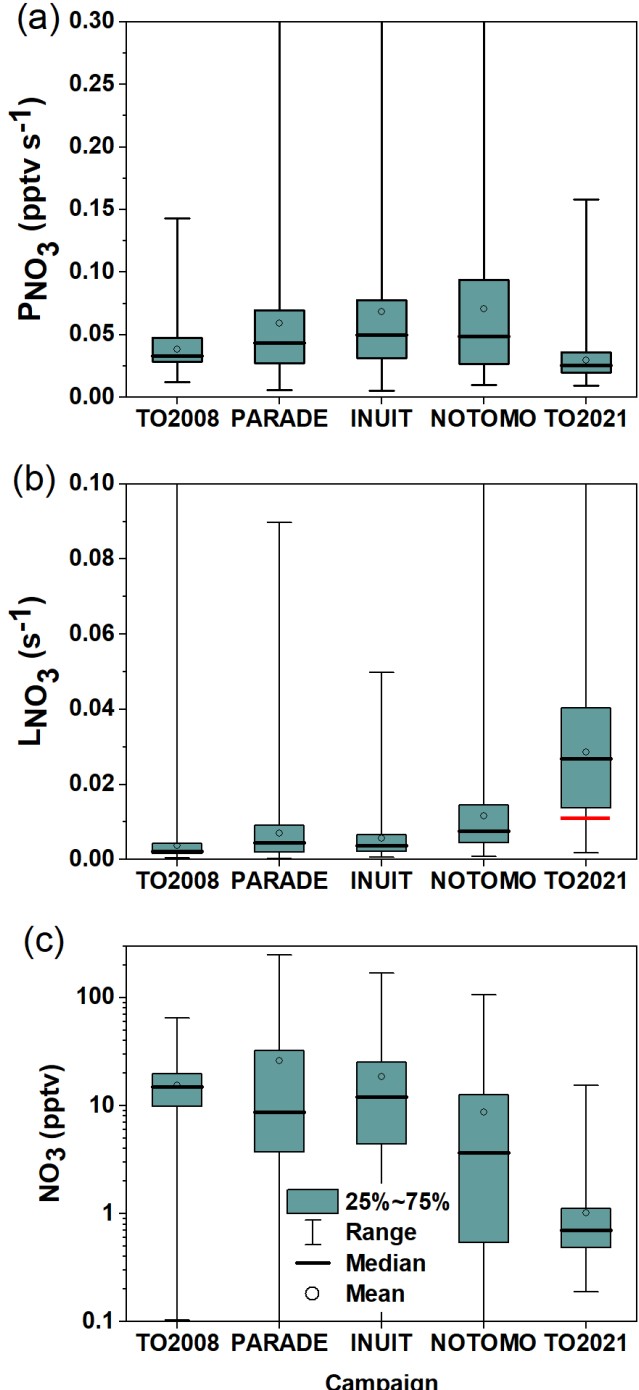

**Figure 13:** $NO_3$ production rates (a), loss rates (b) and mixing ratios (c) measured during 5 campaigns on the summit of Kleiner Feldberg between 2008 and 2021. Boxes represent the range between the first and third quartiles, whiskers denote the full range of values. The red line represents the median of directly measured $k^{NO_3}$ during TO2021 at nighttime.