# Peer review of "Fate of the nitrate radical at the summit of a semi-rural mountain site in Germany assessed with direct reactivity measurements"

_Atmospheric Chemistry and Physics, 2022_

## Author Comment (AC1)

**Reply to Anonymous Referee #1**

*In the following, the referee's comments are reproduced (black) along with our replies (blue) and changes made to the text (red) in the revised manuscript. Line numbers refer to those in the preprint.*

The present work provides an evaluation of $NO_3$ radical fates in a semi-rural site thanks to direct $NO_3$ reactivity measurements during the TO2021 campaign in summer 2021. A Flow-Tube Cavity Ring Down Spectrometer (FT-CRDS) setup was used to measure the $NO_3$ total reactivity and to estimate the contribution of BVOCs to this total reactivity. During this campaign, a number of other relevant measurements ($NO_x$, $O_3$, actinic flux, VOCs, …) were performed to allow for a comprehensive interpretation of the observations.

This study is fully relevant and the FT-CRDS is a very interesting technique to better understand the role of $NO_3$ in the night-time chemistry. The paper is well written and provides detailed information on the experimental setup as well as a very thorough interpretation of the observations, and it is very much appreciable. In general, the scientific quality of this work is very good and once the authors have addressed the following minor points, I would be happy to recommend its publication in ACP.

We thank the Referee for this positive evaluation of our manuscript.

Specific comments:

61: more detailed reactions should be provided to better explain the formation of $RONO_2$ from VOC+$NO_3$ reactions

R10 and R11 do not intend to do more than indicate that $NO_3$ initiated oxidation of unsaturated VOCs can lead to $RONO_2$ and SOA. Given the large number of different VOCs involved and the complexity of the chemistry, a detailed description of the reactions leading to $RONO_2$ and SOA is not waranted here. For this we provide several references. We have replaced VOC with R=R to indicate that unsaturated VOCs are involved:

$NO_3 + R=R (+ O_2) \rightarrow\rightarrow RONO_2$

175: it is not clear why the PTR-MS (VOCUS) was not calibrated with the standard used for the other PTR-MS (Ionicon). Could the authors provide an explanation?

Fragmentation patterns in the VOCUS PTR-MS are not yet completely characterized. We now write:

Fragmentation patterns in the VOCUS PTR-MS are not yet completely characterized and first results (using the same gas standard as for the Ionicon PTR8000) suggest that different monoterpenes fragment differently on several masses in the VOCUS instrument, which impedes calibration of the monoterpene data based on the alpha-pinene standard.

170 and 290: the authors mention that sesquiterpenes were measured but no data/plot have been provided. If available, please provide these data in Figure 3 or in SI. Were sesquiterpenes monitored during previous campaign using GC techniques? Even though sesquiterpenes mixing ratios are expected to be very low, they are suspected to significantly contribute to $NO_3$ fate due their high reactivity. More information about the role of sesquiterpenes on $NO_3$ loss would be useful.

The sesquiterpene data is already provided in the SI in Fig. S4b. We now explicitly refer to it in L176:

In order to extend data availability, the VOCUS data for isoprene, monoterpenes and sesquiterpenes was scaled to that of the PTR8000 data set (which suffered from less

fragmentation, thus associated with less uncertainty) applying constant factors during the common time period (see Fig. S4b in the Supplement).

Unfortunately, no speciated measurements of sesquiterpenes on the KF are available, which is why we used the rate coefficient of beta-caryophyllene to account for their contribution to $NO_3$ losses:

To calculate $NO_3$ loss rates resulting from its reaction with sesquiterpenes, we used the IUPAC-recommended rate coefficient for $NO_3$ + β-caryophyllene since this was the compound used to calibrate the PTR8000. Speciated sesquiterpene measurements on the KF are not available.

The authors do not consider the role of $RO_2$ radicals in the $NO_3$ Do they consider that it is negligible? Previous field studies (e.g. Sommariva et al, 2007) suggest that reactivity with $RO_2$ radicals is not negligible even though $RO_2$ concentrations are very low. This point should be discussed and arguments should be provided for not considering these reactions.

A maximum $RO_x$ mixing ratio of 20 pptv has been reported on the KF (Handisides, 2001), which would correspond to an $NO_3$ loss rate of 0.001 $s^{-1}$, which is insignificant compared to other losses. We now mention this in L317:

Losses due to reaction with $RO_2$ radicals on this site are expected to be insignificant. Taking the average maximum $RO_x$ mixing ratio of 20 pptv as measured by (Handisides, 2001) and the corresponding rate coefficient (IUPAC, 2022) results in an $NO_3$ loss rate of 0.001 $s^{-1}$, which is insignificant compared to the other loss rates mentioned above.

324: the authors cannot conclude that $NO_3$ significantly contributes to the BVOC oxidation during the daytime just because the reactions with BVOCs have been shown to significantly contribute to the $NO_3$ total reactivity. To state that, the BVOCs lifetimes due to $NO_3$ oxidation should be compared to those estimated for OH chemistry (using typical OH concentrations).

We did, in fact, NOT state that $NO_3$ contributes significantly to BVOC oxidation during the day. We wrote *"This underlines that $NO_3$, often considered to be important only at night, also contributes to the oxidation of BVOC during the day and thus potentially to the formation of organic nitrates (in competition to OH- and O3-initiated oxidation) throughout the diel cycle…"* , which is certainly true. However, as the referee has raised this point, we now present a simple calculation to roughly indicate the contribution of $NO_3$ to the daytime loss of a specific BVOC:

Assuming noon mixing ratios of 0.1 pptv $NO_3$ (see Fig. 9 below), 42 ppbv $O_3$ (see Fig. 3) and $10^6$ molecules $cm^{-3}$ OH (Lelieveld et al., 2016) and taking evaluated rate coefficients (IUPAC, 2022) the loss-rate constant of limonene towards these three oxidants would be 2.71 x $10^{-5}$ $s^{-1}$, 2.08 x $10^{-4}$ $s^{-1}$ and 1.65 x $10^{-4}$ $s^{-1}$. $NO_3$ would thus contribute ca. 7 % to the noon-time oxidation of limonene.

340-387: A very detailed discussion on the NOx budget is provided but does not seems to be fully relevant here, in my opinion. It's not clear for me what the authors want to demonstrate here. As a minimum, this should be provided with a clearer objective and in a dedicated section. It has nothing to do in the section "fractional contribution of VOCs to $NO_3$ losses".

NO mixing ratios of several tens of pptv represent a significant loss for $NO_3$ and thus deserve close scrutiny. The presence of both NO and $O_3$ implies a significant $NO_2$ source, which was not reflected in the measured $NO_2$ mixing ratios, implying additional $NO_2$ losses. We now start a separate section and add an introductory sentence to point this out:

**3.5 Effect of nighttime NO on $NO_x$ budget**

Figure 8 reveals a large night-to-night variability in the NO mixing ratio with minimum values close to the detection limit and maxima > 80 pptv in the presence of several tens of ppbv of $O_3$.

The presence of NO and $O_3$ at the mixing ratios observed implies a significant source of $NO_2$. In the following, we derive the NO emission and $NO_2$ deposition rates required to explain the observed NO, $NO_2$ and $O_3$ mixing ratios.

**References**

Handisides, G. M.: The influence of peroxy radicals on ozone production, PhD thesis, Fachbereich Geowissenschaften, Johann Wolfgang Goethe Universität, Frankfurt am Main, 2001.

IUPAC: Task Group on Atmospheric Chemical Kinetic Data Evaluation, edited by: Ammann, M., Cox, R.A., Crowley, J.N., Herrmann, H., Jenkin, M.E., McNeill, V.F., Mellouki, A., Rossi, M. J., Troe, J. and Wallington, T. J., available at: http://iupac.pole-ether.fr/index.html, last access: 24 April 2022.

Lelieveld, J., Gromov, S., Pozzer, A., and Taraborrelli, D.: Global tropospheric hydroxyl distribution, budget and reactivity, Atmos. Chem. Phys., 16, 12477-12493, 2016.

---

## Author Comment (AC2)

**Reply to Anonymous Referee #2**

*In the following, the referee's comments are reproduced (black) along with our replies (blue) and changes made to the text (red) in the revised manuscript. Line numbers refer to those in the preprint.*

The authors report measurements of $NO_3$ reactivity in a summer campaign at Kleiner Feldberg in Germany. They analyse the measurements in terms of different contributions of reactants to the total $NO_3$ loss and draw conclusions about emissions and losses of nitrogen oxides at that place. Results are compared with results from previous campaigns. The manuscript is overall well written and within the scope of the journal. There are some open questions and simplifications that need to be further explained and discussed before the manuscript can be published.

We thank the Referee for taking the time to evaluate our manuscript and for providing useful comments.

L29: $NO_2$ is only a net ozone production, if emitted as $NO_2$. The majority of net $O_3$ is produced from peroxy radical reactions with NO.

We did NOT state that $NO_2$ is the source of net-ozone production, but simply referred to the fact that (via its photolysis) $NO_2$ is the predominant precursor of $O_3$. However, to avoid confusion, we have modified the sentence accordingly in L29:

Via its dissociation, $NO_2$ is the direct, photochemical precursor of boundary layer ozone ($O_3$, a phytotoxin and cause of respiratory illness) and understanding the processes that remove $NO_x$ (= NO + $NO_2$) is of great importance (Crutzen and Lelieveld, 2001; Lelieveld et al., 2016; Edwards et al., 2017).

L140: What could have been the reason for the higher loss rate in the large tube?

The large flowtube was unused and exposed to ambient air for several months before its deployment in TO2021, so that contamination of the walls might be responsible for the high loss rate. Usually, an extended conditioning period with exposure to high $NO_3$ amounts prior to deployment is necessary. However, this is only speculative and already briefly indicated in L295. We prefer not to elaborate on this point.

L143: It would be good, if numbers for the $NO_3$ concentrations that is used in the reactivity instrument were given and compared to ambient concentrations that are expected.

As indicated in L107, the $NO_3$ mixing ratios provided from the source are 30-60 pptv, while ambient $NO_3$ mixing ratios, based on previous measurements on the KF could vary from zero to several tens of pptv. In L143 we add the following:

During the nighttime, before being mixed with 30-60 pptv synthetic $NO_3$, the air was sampled through a 2 $L$ uncoated glass flask (40 s residence time) that was heated to 35°C. This ensures that ambient $NO_3$ and $N_2O_5$ (at mixing ratios up to several tens of pptv according to previous measurements, see below) does not reach the flow tube to bias the measurement.

L161: Did you test if the calibration gas standard used to calibrate the CLD gave the correct concentration in the CRDS instrument?

The NO mixing ratio of the secondary standard used during TO2021 was re-measured with both the CLD and the CRDS instrument after the campaign, which lead to values of (4.67±0.3) ppmv and (4.88±0.22) ppmv respectively. The measurements thus agree within uncertainties, which is added in L162:

Note that post-campaign quantification of the NO standard with the CLD and the TD-CRDS setup yielded satisfactorily agreeing values of (4.7±0.3) ppmv and (4.9±0.2) ppmv.

L175: Why was the VOCUS PTR-MS not calibrated with the same gas standard as the other PTR instrument? Was the scaling factor that needed to be applied to the VOCUS instrument constant for a specific mass for the period, when both instruments measured together?

Fragmentation patterns in the VOCUS PTR-MS are not yet completely characterized. The scaling factor was constant. We now write:

Fragmentation patterns in the VOCUS PTR-MS are not yet completely characterized and first results (using the same gas standard as for the Ionicon PTR8000) suggest that different monoterpenes fragment differently on several masses in the VOCUS instrument, which impedes calibration of the monoterpene data based on the alpha-pinene standard.

In order to extend data availability, the VOCUS data for isoprene, monoterpenes and sesquiterpenes was therefore scaled to that of the PTR8000 data set (which suffered from less fragmentation, thus associated with less uncertainty) applying constant factors during the common time period (see Fig. S4b in the Supplement).

We also added in the caption of Fig. S4b:

Time-series of monoterpenes signals (m/z = 137.132, upper panel) and sesquiterpene signals (m/z = 205.195, lower panel) from the VOCUS data (red) scaled with a constant factor for each mass to the calibrated data from the PTR8000 setup (black).

L195: How was the zero-value determined of the CLD? How often was this done and how stable was the zero?

Instrumental zero was estimated every two hours together with the calibration. The standard deviation of consecutive zeros served as measure for the mentioned LODs. We add this information in L156:

Calibration (using a dynamically diluted, secondary 5 ppm NO standard) was carried out every 2 hours together with the zero measurement using synthetic air (Westfalen). The LODs for NO and $NO_2$ were derived from the standard deviation ($1\sigma$) of consecutive zeros and were 7 and 10 pptv, respectively, the total measurement uncertainties were 9 and 19 % for NO and $NO_2$.

Fig. 1: Why is there only a limited period of monoterpene measurements shown, if the VOCUS PTR was used to complete the measurements as shown in Fig. S4?

Fig.1 only contained the calibrated monoterpene measurement provided by the PTR8000 instrument. We now add the scaled data from the VOCUS-PTR and mention this in the caption:

Overview of key measurements during the TO2021 campaign with wind direction (WD), temperature (T), sum of monoterpenes ($\Sigma$MT, PTR8000 and scaled VOCUS), wind speed (WS), relative humidity (RH), $NO_3$ photolysis rate coefficient ($J_{NO_3}$). […]

L253: Was the height of the vegetation below the tip of the inlet?

The bush and scrub-like vegetation close to the inlet was below the tip of the inlet. We add this information in L253:

These observations support the presence of a very shallow surface layer with its top located below the tip of the inlet and decoupling of the sampled air from ground-level emissions (i.e. of NO and VOCs). The top of the bush and shrub-like vegetation adjacent to the inlet (within a 20 m radius) was several meters below the top of the inlet.

Fig. 4: It would be useful to indicate the inlet height and the height of the vegetation.

We added a black dashed line in Fig. 4 indicating the inlet height. It is dificult to define an average height of the vegetation on a mountain top populated with trees and shrubs, and we prefer not to indicate a vegetation height in the plot. This information is now appended in the caption of the corresponding figure:

[…] The inlet height is indicated by a black dashed line. […]

L 297: It does not make sense to give 3 counting digits for the fractional distribution, if the accuracy of measurements does not provide this accuracy.

The fractional contributions were taken from a publication (cited) and we prefer to quote the same numbers here.

L289: Can you justify, why you expect the same contributions of monoterpenes like in the other campaign? Seasonality, meteorological conditions, changes in the vegetation may highly impact the mix of emissions. This should be further discussed and not neglected as indicated in the in the text.

The data from PARADE2011 represents the only speciated measurement of terpenes at this site and it is reasonable to use this to guide our analysis. We emphasise that making this assumption is associated with significant uncertainty:

We recall however, that speciated monoterpenes were not measured in TO2021 and the effective rate constant was based on the (non-testable) assumption that the summertime monoterpene composition at this site is the same as in 2011. Seasonal and meteorological variations and changes in vegetation over the years mean that this assumption (and the slope of 1.04) is associated with significant uncertainty.

L290: Why is beta-caryophyllene a suitable proxy for the measurement of the sum of sesquiterpenes?

No speciated sesquiterpene measurements are available for the KF. We chose beta-caryophyllene since this is a dominant sesquiterpene and was the compound used for calibration of the PTR8000 instrument. We mention this on L290:

As speciated measurements of sesquiterpenes are not available, in order to calculate $NO_3$ loss rates resulting from its reaction with all sesquiterpenes, we used the IUPAC-recommended rate coefficient for $NO_3$ + β-caryophyllene. This is often the dominant sesquiterpene measured in air and is also the sesquiterpene used to calibrate the PTR8000.

Fig. 5b: It is not clear, what the grey boxes are.

The grey shaded areas (boxes) indicate the uncertainty associated with the NO3 reactivity measurement. We add this information in the caption of Fig. 5a:

[…] The red- and grey-shaded areas represent the uncertainty associated with $k^{NO_3}$ and $\Sigma k_i[VOC]_i$, respectively. (b) Same as (a) but with a detailed view of the night between the 22$^{nd}$ and 23$^{rd}$ July presented in Fig.2b.

Fig. 5c: Was there no other (unaccounted) NO3 reactivity on average?

No, the directly measured $NO_3$ reactivity was on average slightly lower than $\Sigma k_i[VOC]_i$ (see intercept and slope in Fig. 6 and the new Fig. S4c), which is why latter was used to estimate the contributions as indicated in the corresponding figure caption.

L 294 and Fig. 5a: The figure gives the impression that monoterpene species can explain the NO3 reactivity. However, it would be easier to judge this if the x-scale was wider and/or the time period was split into 2 panels.

We have modified Fig. 5a by adding the contribution of monoterpenes with a solid purple line to allow a better separation from $\Sigma k_i[VOC]_i$. We modified the corresponding figure caption accordingly:

(a) Time-series of $k^{NO_3}$ and $\Sigma k_i[VOC]_i$. Dashed blue line marks the LOD of the $k^{NO_3}$ measurement. The purple line together with the same-coloured shade represents the contribution of monoterpenes. [...]

L 303ff and Fig. 6: It looks as if there are more data points than shown in the figure. The large symbol size and using also black colour for the error bars makes it is hard to see details. What is the correlation coefficient? The distribution is very wide and shows that there are also a high number of points where numbers are not the same. A plot of the time series of the difference between calculated and measured NO3 reactivity could give more insights if this is due to statistically or systematic differences during specific periods of the campaign.

We agree and modified Fig. 6 by omitting the error bars and using a smaller symbol size. The correlation coefficient is 0.8 and suggests a fair correlation. We added to the caption of Fig. 6: For the sake of better clarity, error bars were omitted.

The scatter may originate from changes in the monoterpene composition or the different location of the inlets. A time-series of the difference between measured and calculated NO₃ reactivity is appended as Fig. S4c in the Supplement and indicates the systematic deviations discussed in the main text. We made the following changes in the manuscript in L303ff:

The correlation coefficient of 0.8 indicates a reasonable quality of fit. This is also seen (Fig. S4c in the Supplement) in a time-series showing the difference between $k^{NO_3}$ and $\Sigma k_i[VOC]_i$. The scatter in both plots is likeyl to be caused by changes in the monoterpene composition or the different location of the instruments' inlets.

L314ff and Fig. 7: It should be emphasized / defined that NO3 reactions with VOCs are meant, if you say "fractional contribution F".

The fact that $k^{NO_3}$ is a direct measure for the NO₃ reactivity towards VOCs is emphasized within the text by the sentences right before (L312-313) and after (L320) the definition of F in L314. We modified the caption of the y-axis in Fig. 7 as well as the caption of Fig. 7 to make this clear:

Mean, fractional contribution (F) of $k^{NO_3}$ (i.e. VOC contribution) to the overall NO₃ loss rate over the diel-cycle.

L319: What are the reasons for the increase of the contribution of NO3 + VOC reactions?

The slight increase in F between 08:00 and 16:00 UTC is accompanied by a decrease in NO mixing ratios (Fig. 3f). The sharp increase after 16:00 UTC is caused by both decreasing actinic flux and lower NO mixing ratios (originating from NO₂ photolysis). We now explain this in L319:

The fractional contribution of VOC-induced losses is low at noon (~ 9 %) but increases to ~ 30% in the afternoon at 18:00 UTC due to the decrease in NO levels between 08:00 and 16:00 UTC (Fig. 3f) and to decreasing actinic flux and the associated slowing of both NO₃ and NO₂ photolysis to NO beginning at 16:00 UTC.

L324: Can you give an estimate how the reaction rate of VOCs with OH and O3 were during daytime to support your statement about the importance of NO3 reactions for the oxidation of BVOCs during the day?

As an example, the lifetime of limonene to the major oxidants OH, O₃ and NO₃ were estimated by using their expected or measured average noon concentrations and the corresponding evaluated rate coefficients for the reaction at 298 K. Taking the median noon NO₃ (steady-state, Fig.7) and O₃ mixing ratios of 0.1 pptv and 42 ppbv respectively and assuming an OH concentration of $10^6$ molecules cm⁻³, the loss rate of limonene to NO₃ is 2.71 x $10^{-5}$ s⁻¹, while that to O₃ and OH is 2.08 x $10^{-4}$ s⁻¹ and 1.65 x $10^{-4}$ s⁻¹ respectively. According to this, NO₃ thus contributes with 7 % to daytime oxidation. We thus added in L 324:

Assuming noon mixing ratios of 0.1 pptv $NO_3$ (see Fig. 9), 42 ppbv $O_3$ (see Fig. 3) and $10^6$ molecules $cm^{-3}$ OH (Lelieveld et al., 2016) and taking evaluated rate coefficients (IUPAC, 2017) the lifetime of limonene towards these three oxidants would be $2.71 \times 10^{-5} s^{-1}$, $2.08 \times 10^{-4} s^{-1}$ and $1.65 \times 10^{-4} s^{-1}$. $NO_3$ would thus contribute ca. 7 % to the daytime oxidation of limonene.

L327: Do you want to say that local anthropogenic emissions existed only during daytime? Why would this be the case?

We expect reduced traffic at nighttime and thus less anthropogenic emission of NO. We note the misleading nature of this sentence in L327 and rephrase it:

At nighttime, in the absence of actinic radiation (to convert $NO_2$ to NO) and less local anthropogenic NO emissions due to reduced traffic, NO levels are generally suppressed by reaction with $O_3$.

L339ff: As discussed a bit later, you may expect a strong gradient of NO concentrations with height also within the surface layer due to the rapid reaction with O3 unless the mixing is fast, which may not be expected specifically in the night. Does your estimate of the NO concentration consider such a gradient, if the inlet of the NO instrument is at a certain height? I assume that the NO source from soil would need to be significantly higher, if this is taken into account.

We agree, the calculation assumes homogeneous mixing within the proposed 10 m NBL height. Since no vertical profile measurements of NO were available, this calculation serves as upper limit. A common way to account for poor mixing is to introduce a factor of 2 for the NBL height dependent deposition velocity, which derives from the assumption of a linear increase with height. We clarify this by adding the following in L340:

Note however, that this estimation assumes a mixed layer. Assuming a linear gradient in NO mixing ratios with height, the NO emission rates at ground level would be a factor of two higher (Shepson et al., 1992; Fischer et al., 2019).

L366ff: Does the model also include O3, NO3, N2O5 deposition? If so, this should be mentioned, if not it needs to be justified, why no deposition is assumed. These loss processes would all contribute to the loss of odd oxygen, and it may not be easy to distinguish between the loss for the different species. How can you justify that a 0-D box model is applicable for modelling measurements made close to the ground in night-time conditions, when mixing is poor?

No, heterogeneous losses of $O_3$, $NO_3$ and $N_2O_5$ are not included. Deposition of $NO_3$ and $N_2O_5$ were found to be insignificant compared to the gas-phase $NO_3$ loss rates during previous measurements on this site (Tab. 1). $O_3$ deposition is not considered because $O_3$ mixing ratios were constrained to measured values (already mentioned in L368). We do not wish to imply that the simple, zero-dimensional model provides an accurate simulation of the BL processes, but gives us an oder-of-magnitude idea of the $NO_2$ loss needed to explain the results. We emphasize these issues in L371:

Heterogeneous losses of $NO_3$ and $N_2O_5$ were not considered since these were found to be insignificant compared to gas-phase losses (of $NO_3$) during previous campaigns on the KF (see below, Tab. 1). Furthermore, note that vertical gradients are not considered by this simulation which aims to provide a ball-park value for the $NO_2$ loss term needed to explain its mixing ratios in the presence of a known production rate.

L456ff and Table 1: It would be good to see the comparison of NO measurements and have this discussed in more detail the text. If the explanation for the high NO in this campaign is soil emissions, what for example could be reason, why this was not observed in the other campaigns?

We agree and have added the range of measured NO mixing ratios during TO2008 and PARADE in Table 1. During these campaigns, nighttime NO mixing were above LOD on some nights, but usually did not exceed 30 pptv. The soil emission strength is dependent on various parameters. A characteristic feature of TO2021 compared to the others were wet conditions throughout almost the whole period which might lead to soil moisture ranges favouring higher NO emission rates. We add this in L459:

As shown in Tab. 1 nighttime NO mixing during TO2008 and PARADE usually did not exceed 30 pptv. Several parameters impact the NO emission rate of soils (Pilegaard, 2013) and since TO2021 was exceptionally wet compared to previous campaigns, a greater soil water content may have favoured high NO emissions in TO2021.

Technical comments:
General technical comment: It makes it easier to read and follow the manuscript if less abbreviations are used in the text.
We generally agree to that statement. The abbreviations used are however common or necessary to avoid frequent repetition of long text-segments (e.g. campaign names).

L36: "OH reactions being most important" instead of "OH reactions most important"
Correction made.

L61: subscript RONO_2
Correction made.

Fig. 2: You may need to increase the font size if these figures become 1-column figures.
Font sizes increased.

L312: comma missing after (j_NO3).
Correction made.

L401: Units of productions rates are pptv s-1 and not s-1.
Correction made.

**References**

Fischer, H., Axinte, R., Bozem, H., Crowley, J. N., Ernest, C., Gilge, S., Hafermann, S., Harder, H., Hens, K., Janssen, R. H. H., Konigstedt, R., Kubistin, D., Mallik, C., Martinez, M., Novelli, A., Parchatka, U., Plass-Dulmer, C., Pozzer, A., Regelin, E., Reiffs, A., Schmidt, T., Schuladen, J., and Lelieveld, J.: Diurnal variability, photochemical production and loss processes of hydrogen peroxide in the boundary layer over Europe, Atmos. Chem. Phys., 19, 11953-11968, doi:10.5194/acp-19-11953-2019, 2019.

IUPAC: Task Group on Atmospheric Chemical Kinetic Data Evaluation, edited by: Ammann, M., Cox, R.A., Crowley, J.N., Herrmann, H., Jenkin, M.E., McNeill, V.F., Mellouki, A., Rossi, M. J., Troe, J. and Wallington, T. J., available at: http://iupac.pole-ether.fr/index.html, last access: 25 April 2022.

Lelieveld, J., Gromov, S., Pozzer, A., and Taraborrelli, D.: Global tropospheric hydroxyl distribution, budget and reactivity, Atmos. Chem. Phys., 16, 12477-12493, 2016.

Pilegaard, K.: Processes regulating nitric oxide emissions from soils, Philos. Trans. R. Soc., B, 368, 1-8, doi:10.1098/rstb.2013.0126, 2013.

Shepson, P. B., Bottenheim, J. W., Hastie, D. R., and Venkatram, A.: Determination of the relative ozone and PAN deposition velocities at night, Geophys. Res. Lett., 19, 1121-1124, doi:10.1029/92gl01118, 1992.